# Differential methods for assessing sensitivity in biological models

**Rachel Mester**[1]*, **Alfonso Landeros**[1], **Chris Rackauckas**[2,3,4], **Kenneth Lange**[1,5,6]

**1** Department of Computational Medicine, University of California Los Angeles, Los Angeles, California, United States of America, **2** Computer Science and Artificial Intelligence Laboratory, Massachusetts Institute of Technology, Cambridge, Massachusetts, United States of America, **3** Pumas-AI, Annapolis, Maryland, United States of America, **4** Julia Computing, Cambridge, Massachusetts, United States of America, **5** Department of Human Genetics, University of California Los Angeles, Los Angeles, California, United States of America, **6** Department of Statistics, University of California Los Angeles, Los Angeles, California, United States of America

* rmester@ucla.edu

**Data Availability Statement:** There is no primary data in the paper; all materials are available at https://github.com/rachelmester/ SensitivityAnalysis.

## Abstract

Differential sensitivity analysis is indispensable in fitting parameters, understanding uncertainty, and forecasting the results of both thought and lab experiments. Although there are many methods currently available for performing differential sensitivity analysis of biological models, it can be difficult to determine which method is best suited for a particular model. In this paper, we explain a variety of differential sensitivity methods and assess their value in some typical biological models. First, we explain the mathematical basis for three numerical methods: adjoint sensitivity analysis, complex perturbation sensitivity analysis, and forward mode sensitivity analysis. We then carry out four instructive case studies. (a) The CARRGO model for tumor-immune interaction highlights the additional information that differential sensitivity analysis provides beyond traditional naive sensitivity methods, (b) the deterministic SIR model demonstrates the value of using second-order sensitivity in refining model predictions, (c) the stochastic SIR model shows how differential sensitivity can be attacked in stochastic modeling, and (d) a discrete birth-death-migration model illustrates how the complex perturbation method of differential sensitivity can be generalized to a broader range of biological models. Finally, we compare the speed, accuracy, and ease of use of these methods. We find that forward mode automatic differentiation has the quickest computational time, while the complex perturbation method is the simplest to implement and the most generalizable.

## Author summary

Over the past few decades, mathematical modeling has become an indispensable tool in the biologist's toolbox. From deterministic to stochastic to statistical models, computational modeling is ubiquitous in almost every field of biology. Because model parameter estimates are often noisy or depend on poorly understood interactions, it is crucial to examine how both quantitative and qualitative predictions change as parameter estimates

**Funding:** RM research supported by NIH grant T32HG002536. KL research supported by USPHS grants HG006139 and GMS141798. CR research supported by the National Science Foundation under grant no. OAC-1835443, grant no. SII-2029670, grant no. ECCS-2029670, grant no. OAC-2103804, and grant no. PHY-2021825, the U.S. Agency for International Development through Penn State for grant no. S002283-USAID, the Advanced Research Projects Agency-Energy (ARPA-E), U.S. Department of Energy, under Award no. DE-AR0001211 and DE-AR0001222, The Research Council of Norway and Equinor ASA through Research Council project "308817 - Digital wells for optimal production and drainage", the United States Air Force Research Laboratory and the U.S. Air Force Artificial Intelligence Accelerator under Cooperative Agreement Number FA8750-19-2-1000. The views and opinions of authors expressed herein do not necessarily state or reflect those of the United States Government or any agency thereof. The funders had no role in study design, data collection, and analysis.

**Competing interests:** The authors have declared that no competing interests exist.

change, especially as the number of parameters increases. Sensitivity analysis is the process of understanding how a model's behavior depends on parameter values. Sensitivity analysis simultaneously quantifies prediction certainty and clarifies the underlying biological mechanisms that drive computational models. While sensitivity analysis is universally recognized to be an important step in modeling, it is often unclear how to best leverage the available differential sensitivity methods. In this manuscript we explain and compare various differential sensitivity methods in the hope that best practices will be widely adopted. We stress the relative advantages of existing software and their limitations. We also present a new numerical technique for computing differential sensitivity.

This is a *PLOS Computational Biology* Methods paper.

## 1 Introduction

In many mathematical models underlying parameters are poorly specified. This problem is particularly acute in biological and biomedical models. Model predictions can have profound implications for scientific understanding, further experimentation, and even public-policy decisions. For instance, in an epidemic some model parameters can be tweaked by societal or scientific interventions to drive infection levels down. Differential sensitivity can inform medical judgement about the steps to take with the greatest impact at the least cost. Similar considerations apply in economic modeling. Additionally, parameter estimation for model fitting usually involves differential sensitivity through maximum likelihood or least squares criteria. These optimization techniques depend heavily on gradients and Hessians with respect to parameters. While some parameter estimation methods rely on Bayesian computational techniques [1] rather than gradients, these techniques tend to scale poorly as the number of model parameters increases. A common way to alleviate the poor scaling of Bayesian inference is Hamiltonian Monte Carlo [2], which itself requires gradient calculations. Techniques for assessing sensitivity of stochastic models often rely on the gradient-dependent Fisher information matrix of the model, which is the basis for a variety of multi-step local sensitivity analysis techniques for discrete stochastic models [3].

Calculation of gradients and Hessians of a model can also be important in other steps of the scientific process. For example, iterative model development [4] involves using the Fisher information matrix to inform experimental design. Extended Kalman filtering [5] incorporates differential sensitivity into model construction. Regardless of the method, parameter estimation is an important step in fitting a biological model, and the success of this step strongly impacts the ultimate utility of the model. Understanding the uses and limitations of differential sensitivity can aid in determining the identifiability of model parameters, how sensitive they are to experimental error or measurement noise, and the overall importance of their existence in the model. Finally, it is worth noting that while local sensitivity analysis is the focus of this manuscript, global sensitivity analysis often relies on local differential sensitivity estimates to inform optimal stepsizes in regional searching [6] or to resolve inconsistencies that arise when local sensitivity is non-monotonic [7].

In any case it is imperative to know how sensitive model predictions are to changes in parameter values. Unfortunately, assessment of model sensitivity can be time consuming, computationally intensive, inaccurate, and simply confusing. Most models are nonlinear and

resistant to exact mathematical analysis. Understanding their behavior is only approachable by solving differential equations or intensive and noisy simulations. Sensitivity analysis is often conducted over an entire bundle of neighboring parameters to capture interactions. If the parameter space is large or high-dimensional, it is often unclear how to choose representative points from this bundle. Faced with this dilemma, it is common for modelers to fall back on varying just one or two parameters at a time. Model predictions also often take the form of time trajectories. In this setting, sensitivity analysis is based on lower and upper trajectories bounding the behavior of the dynamical system.

The differential sensitivity of a model quantity is measured by its gradient with respect to the underlying parameters at their estimated values. The existing literature on differential sensitivity is summarized in the modern references [8,9]. There are a variety of software packages that evaluate parameter sensitivity. For example, the Julia software DifferentialEquations.jl [10] makes sensitivity analysis routine for many problems. Additionally, PESTO [11] is a current Matlab toolbox for parameter estimation that uses adjoint sensitivities implemented as part of the CVODES method from SUNDIALS [12]. Although the physical sciences have widely adopted the method of differential sensitivity [13,14], the papers and software generally focus on a single sensitivity analysis method rather than a comparison of the various approaches. This singular focus leaves open many questions when biologists conduct sensitivity analyses. Should the continuous sensitivity equations be used, or would automatic differentiation of solvers be more efficient on biological models? On the types of models biologists generally explore, would implicit parallelism within the sensitivity equations be beneficial, or would the overhead cost of thread spawning overrule any benefits? How close do simpler methods based on complex perturbation get to these techniques? The purpose of the current paper is to explore these questions on a variety of models of interest to computational biologists.

In the current paper we also suggest an accurate method of approximating gradients that compares favorably against forward automatic differentiation techniques, provided a model involves analytic functions without discontinuities, maxima, minima, absolute values, or any other excursion outside the universe of analytic functions. In the sections immediately following, we summarize known theory, including the important adjoint method for computing the sensitivity of functions of solutions [13, 14]. Then we illustrate sensitivity analysis for a few deterministic models and a few stochastic models. Our exposition includes some straightforward Julia code that readers can adapt to their own sensitivity needs. These examples are followed by an evaluation of the accuracy and speed of the suggested numerical methods. The concluding discussion summarizes our experience, indicates limitations of the methods, and suggests new potential applications.

For the record, here are some notational conventions used throughout the paper. All functions that we differentiate have real or real-vector arguments and real or real-vector values. All vectors and matrices appear in boldface. The superscript indicates a vector or matrix transpose. For a smooth real-valued function $f(\mathbf{x})$, we write its gradient (column vector of partial derivatives) as $\nabla f(\mathbf{x})$ and its differential (row vector of partial derivatives) as $df(\mathbf{x}) = \nabla f(\mathbf{x})^T$. If $g(\mathbf{x})$ is vector-valued with $i$th component $g_i(\mathbf{x})$, then the differential (Jacobi matrix) $dg(\mathbf{x})$ has $i$th row $dg_i(\mathbf{x})$. The chain rule is expressed as the equality $d[f \circ g(\mathbf{x})] = df[g(\mathbf{x})]dg(\mathbf{x})$ of differentials. The transpose (adjoint) form of the chain rule is $\nabla f \circ g(\mathbf{x}) = dg(\mathbf{x})^T \nabla f[g(\mathbf{x})]$. For a twice-differentiable function, the second differential (Hessian matrix) $d^2 f(\mathbf{x}) = d\nabla f(\mathbf{x})$ is the differential of the gradient. Finally, $i$ will denote $\sqrt{-1}$.

## 2 Methods for computing sensitivity

### 2.1 Forward method

S3 Appendix briefly discusses sensitivity analysis for the linear constant coefficient system $\frac{d}{dt}\mathbf{x}(t) = \mathbf{A}(\boldsymbol{\beta})\mathbf{x}(t)$ of ordinary differential equations (ODEs). Sensitivity of the nonlinear system $\frac{d}{dt}\mathbf{x}(t, \boldsymbol{\beta}) = f[\mathbf{x}(t), \boldsymbol{\beta}]$ can be evaluated by differentiating the original ODE with respect to $\beta_j$, interchanging the order of differentiation, and numerically integrating the system

$$\frac{d}{dt}\frac{\partial}{\partial \beta_j}\mathbf{x}(t, \boldsymbol{\beta}) \quad = \quad \frac{\partial}{\partial \beta_j}f[\mathbf{x}(t), \boldsymbol{\beta}] + d_x f[\mathbf{x}(t), \boldsymbol{\beta}]\frac{\partial}{\partial \beta_j}\mathbf{x}(t, \boldsymbol{\beta}).$$

This formulation of the problem depends on knowing $\mathbf{x}(t, \boldsymbol{\beta})$. In practice, one solves the system

$$\frac{d}{dt}\begin{bmatrix} \mathbf{x}(t, \boldsymbol{\beta}) \\ \nabla_{\boldsymbol{\beta}}\mathbf{x}(t, \boldsymbol{\beta}) \end{bmatrix} \quad = \quad \begin{pmatrix} f[\mathbf{x}(t), \boldsymbol{\beta}] \\ \nabla_{\boldsymbol{\beta}}f[\mathbf{x}(t), \boldsymbol{\beta}] + d_{\boldsymbol{\beta}}\mathbf{x}(t, \boldsymbol{\beta})^T\nabla_x f[\mathbf{x}(t), \boldsymbol{\beta}] \end{pmatrix} \tag{1}$$

jointly, where $d_{\boldsymbol{\beta}}\mathbf{x}[t, \boldsymbol{\beta}]$ is the Jacobi matrix of $\mathbf{x}(t, \boldsymbol{\beta})$ with respect to $\boldsymbol{\beta}$. This is commonly referred to as forward sensitivity analysis and is carried out by software suites such as DifferentialEquations.jl and SUNDIALS CVODES [12]. We note that a common implementation of sensitivity analysis is to base calculations on directional derivatives. Thus, the directional derivative

$$d_{\boldsymbol{\beta}}\mathbf{x}(t, \boldsymbol{\beta})^T\nabla_x f[\mathbf{x}(t), \boldsymbol{\beta}] \quad = \quad \lim_{\epsilon \to 0}\frac{f\{\mathbf{x}(t) + \epsilon\nabla_x f[\mathbf{x}(t), \boldsymbol{\beta}], \boldsymbol{\beta}\} - f[\mathbf{x}(t), \boldsymbol{\beta}]}{\epsilon}$$

version of the forward method allows one to evolve dynamical systems without ever computing full Jacobians. The forward method can also be applied when quantities of interest are defined recursively.

### 2.2 Adjoint methods

The adjoint method is incorporated in the biological parameter estimation software PESTO through CVODES [12]. This method [8,9] is defined directly on a function $g[x(\boldsymbol{\beta}), \boldsymbol{\beta}]$ of the solution of the ODE. The adjoint method introduces a Lagrange multiplier $\lambda(\boldsymbol{\beta})$, numerically solves the ODE system forward in time over $[t_0, t_n]$, then solves the system

$$d_{\boldsymbol{\beta}}\lambda(\boldsymbol{\beta}) \quad = \quad d_x f[\mathbf{x}(\boldsymbol{\beta}), \boldsymbol{\beta}]\lambda(\boldsymbol{\beta}) + d_{\boldsymbol{\beta}}g[\mathbf{x}(\boldsymbol{\beta}), \boldsymbol{\beta}],$$

for $\lambda(\boldsymbol{\beta})$ in reverse time, and finally uses the introduced parameter to compute derivatives via

$$d_{\boldsymbol{\beta}}g[x(\boldsymbol{\beta}), \boldsymbol{\beta}] \quad = \quad \int_{t_0}^{t_n} \lambda(t, \boldsymbol{\beta})d_{\boldsymbol{\beta}}\mathbf{x}(t, \boldsymbol{\beta})dt.$$

The second and third stages are commonly combined by appending the last equation to the set of ODEs being solved in reverse. This tactic achieves a lower computational complexity than other techniques, which require solving an $n$-dimensional ODE system $p$ times for $p$ parameters. In contrast, the adjoint method solves an $n$-dimensional ODE forwards and then solves an $n$-dimensional and a $p$-dimensional system in reverse, changing the computational complexity from $\mathcal{O}(np)$ to $\mathcal{O}(n + p)$. Whether such asymptotic cost advantages lead to more efficiency on practical models is precisely one of the points studied in this paper.

Alternatively, one can find the partial derivatives using finite differences. The simplest method here is to compute a slightly perturbed trajectory $\mathbf{x}(t, \boldsymbol{\beta}+\Delta\mathbf{v})$ and form the forward

differences

$$\frac{\mathbf{x}(t, \boldsymbol{\beta} + \Delta\mathbf{v}) - \mathbf{x}(t, \boldsymbol{\beta})}{\Delta}$$

at all specified time points as approximations to the forward directional derivatives of $\mathbf{x}(t, \boldsymbol{\beta})$ in the direction $\mathbf{v}$. Choosing $\mathbf{v}$ to be unit vectors along each coordinate axis gives ordinary partial derivatives. The accuracy of this crude method suffers from round-off error in subtracting two nearly equal function values. These round-off errors are in addition to the usual errors committed in integrating the differential equation numerically. Round-off errors can be ameliorated by using central differences

$$\frac{\mathbf{x}\left(t, \boldsymbol{\beta} + \frac{\Delta}{2}\mathbf{v}\right) - \mathbf{x}\left(t, \boldsymbol{\beta} - \frac{\Delta}{2}\mathbf{v}\right)}{\Delta}$$

rather than forward differences. However, the central difference method requires twice the number of computations as the forward difference method. Thus, the choice of a difference method depends on prioritization of accuracy versus computational efficiency. In small models, computational efficiency may be less of a priority, in which case central difference methods are preferred.

## 2.3 Complex perturbation methods

There is a far more accurate way of computing model sensitivity when the function $f[\mathbf{x}, \boldsymbol{\beta}]$ defining the ODE is analytic in the parameter vector $\boldsymbol{\beta}$ [15]. An analytic function can be expanded in a locally convergent power series around every point of its domain. This implies that the trajectory $\mathbf{x}(t, \boldsymbol{\beta})$ is also analytic in $\boldsymbol{\beta}$. For a real analytic function $g(\beta)$ of a single variable $\beta$, the derivative approximation

$$g\prime(\beta) \quad = \quad \frac{\text{Imag}\, g(\beta + \Delta i)}{\Delta} + O(\Delta^2)$$

in the complex plane avoids roundoff and is highly accurate for $\Delta > 0$ very small [16,17]. Thus, in calculating a directional derivative of $\mathbf{x}(t, \boldsymbol{\beta})$, it suffices to (a) solve the governing ODE $\frac{d}{dt}\mathbf{x}(t, \boldsymbol{\beta}) = f[\mathbf{x}(t), \boldsymbol{\beta}]$ with $\boldsymbol{\beta} + \Delta i\mathbf{v}$ replacing $\boldsymbol{\beta}$, (b) take the imaginary part of the result, and (c) divide by $\Delta$. To make these calculations feasible, the computer language implementing the calculations should support complex arithmetic and ideally have an automatic dispatching mechanism so that only one implementation of each function is required. In contrast to numerical integration of the joint system (Eq 1), the complex perturbation method is much more simply parallelizable across parameters.

The following straightforward Julia routine for computing sensitivities

```
function differential(f::F, p, Δ) where F
    fvalue = real(f(p)) # function value
    df = zeros(length(fvalue), length(p)) # states x parameters
    pworker = [map(complex, p) for _ in 1:Threads.nthreads()]
    Threads.@threads for j = 1:length(p)
        _p = pworker[Threads.threadid()] # thread worker array
        _p[j] = _p[j] + Δ * im # perturb parameter
        fj = f(_p) # compute perturbed function value
        _p[j] = complex(real(_p[j]), 0.0) # reset parameter
        df[:,j]. = imag(fj)./ Δ # fill in jth partial
    end
    return (fvalue, df)
end
```

takes advantage of the simplicity of multithreading the complex perturbation method by parameter. This function requires a function $f(\mathbf{p})\colon \mathbb{R}^n \mapsto \mathbb{R}^m$ of a real vector $\mathbf{p}$ declared as complex. The perturbation scalar $\Delta$ should be small and real, say $10^{-10}$ to $10^{-12}$ in double precision. If the parameters $p_j$ vary widely in magnitude, then a good heuristic is to perturb $p_j$ by $p_j di$ instead of $di$. The returned value df is an $m \times n$ real matrix. The Julia commands real and complex effect conversions between real and complex numbers, and Julia substitutes im for $i = \sqrt{-1}$. We will employ these functions later in some case studies.

A recent extension [18] of the complex perturbation method facilitates accurate approximation of second derivatives. The relevant formula is

$$\frac{\partial^2}{\partial \beta_j^2} g(\boldsymbol{\beta}) \;=\; \frac{\operatorname{Imag}[g(\boldsymbol{\beta} + e^{\pi i/4}\Delta \mathbf{e}_j) + g(\boldsymbol{\beta} - e^{\pi i/4}\Delta \mathbf{e}_j)]}{\Delta^2} + O(\Delta^4), \tag{2}$$

where $e^{\pi i/4} = (1+i)/\sqrt{2}$. Roundoff errors can now occur but are usually manageable. Here we present a novel result for how to extend the complex perturbation method to approximate mixed partials. Our derivation is condensed into the following equations

$$
\begin{aligned}
\Delta g[\mathbf{x} + e^{\pi i/4}(\mathbf{e}_j + \mathbf{e}_k)] \;\approx\; & \; g(\mathbf{x}) + e^{\pi i/4} dg(\mathbf{x})\Delta(\mathbf{e}_j + \mathbf{e}_k) \\
& + \frac{i}{2}\Delta(\mathbf{e}_j + \mathbf{e}_k)^\top d^2 g(\mathbf{x})\Delta\left(\mathbf{e}_j + \mathbf{e}_k\right) \\
& + \frac{e^{\pi 3/4}}{6} d^3 g\left[\mathbf{x}; \Delta^3(\mathbf{e}_j + \mathbf{e}_k)^3\right] \\
g[\mathbf{x} - e^{\pi i/4}\Delta(\mathbf{e}_j + \mathbf{e}_k)] \;\approx\; & \; g(\mathbf{x}) - e^{\pi i/4} dg(\mathbf{x})\Delta(\mathbf{e}_j + \mathbf{e}_k) \\
& + \frac{i}{2}\Delta(\mathbf{e}_j + \mathbf{e}_k)^\top d^2 g(\mathbf{x})\Delta\left(\mathbf{e}_j + \mathbf{e}_k\right) \\
& - \frac{e^{\pi 3/4}}{6} d^3 g\left[\mathbf{x}; \Delta^3(\mathbf{e}_j + \mathbf{e}_k)^3\right].
\end{aligned}
$$

This approximation is accurate to order $O(\Delta^6)$ and allows us to infer that

$$
\begin{aligned}
\frac{\operatorname{Imag} g[\boldsymbol{x} + e^{\pi i/4\Delta}(\boldsymbol{e}_j + \boldsymbol{e}_k)] + g[\boldsymbol{x} - e^{\pi i/4\Delta}(\boldsymbol{e}_j + \boldsymbol{e}_k)]}{\Delta^2} \;=\; & \\
(\boldsymbol{e}_j + \boldsymbol{e}_k)^\top d^2 g(\boldsymbol{x})(\boldsymbol{e}_j + \boldsymbol{e}_k) + O(\Delta^4) \;=\; & \\
\frac{\partial^2}{\partial \beta_j^2} g(\boldsymbol{\beta}) + \frac{\partial^2}{\partial \beta_k^2} g(\boldsymbol{\beta}) + 2\frac{\partial^2}{\partial \beta_j \partial \beta_k} g(\boldsymbol{\beta}) + O(\Delta^4) &
\end{aligned}
\tag{3}
$$

Since we can approximate $\frac{\partial^2}{\partial \beta_j^2} g(\boldsymbol{\beta})$ and $\frac{\partial^2}{\partial \beta_k^2} g(\boldsymbol{\beta})$, we can now approximate $\frac{\partial^2}{\partial \beta_j \partial \beta_k} g(\boldsymbol{\beta})$ to order $O(\Delta^4)$. These approximations are derived in S1 Appendix.

The Julia code for computing second derivatives

```
function hessian(f::F, p, Δ) where F
    d2f = zeros(length(p), length(p)) # hessian
    dp = Δ * (1.0 + 1.0 * im) / sqrt(2)
    for j = 1:length(p) # compute diagonal entries of d2f
        p[j] = p[j] + dp
        fplus = f(p)
        p[j] = p[j] - 2 * dp
        fminus = f(p)
        p[j] = complex(real(p[j]), 0.0) # reset parameter
        d2f[j, j] = imag(fplus + fminus) / Δ^2
    end
```

```
         for j = 2:length(p) # compute off diagonal entries
             for k = 1:(j−1)
                 (p[j], p[k]) = (p[j] + dp, p[k] + dp)
                 fplus = f(p)
                 (p[j], p[k]) = (p[j] - 2 * dp, p[k] - 2 * dp)
                 fminus = f(p)
                 (p[j], p[k]) = (complex(real(p[j]), 0.0), complex(real(p
  [k]), 0.0))
                 d2f[j, k] = imag(fplus + fminus) / Δ^2
                 d2f[j, k] = (d2f[j, k]−d2f[j, j]−d2f[k, k]) / 2
                 d2f[k, j] = d2f[j, k]
             end
         end
         return d2f
 end
```

operates on a scalar-valued function $f(u)$ of a real vector $\mathbf{p}$ declared as complex. The second-order complex perturbation method can also be multithreaded by parameter, provided the unmixed second partials are computed prior to the mixed ones. Because roundoff error is now a concern, the perturbation scalar $\Delta$ should be in the range $10^{-3}$ to $10^{-6}$ in double precision. The returned value $d^2f$ is a symmetric matrix.

## 2.4 Automatic differentiation

Another technique one can use to calculate the derivatives of model solutions is to differentiate the numerical algorithm that calculates the solution. This can be done with computational tools collectively known as automatic differentiation [19]. Forward mode automatic differentiation is performed by carrying forward Jacobian-vector products at each successive calculation. This is accomplished by defining higher-dimensional numbers, known as dual numbers [20], coupled to primitive functions $f(\mathbf{x})$ through the action

$$f(\mathbf{a} + \mathbf{b}\epsilon) \;=\; f(\mathbf{a}) + \epsilon df(\mathbf{a})\mathbf{b}.$$

Here $\epsilon$ is a dimensional marker, similar to the complex $i$, which is a two-dimensional number. For a composite function $f = f_2 \circ f_1$, the chain rule is $df(\mathbf{a})\mathbf{b} = df_2[f_1(\mathbf{a})]df_1(\mathbf{a})\mathbf{b}$. The $i$th column of the Jacobian appears in the expression $f(\mathbf{x} + \mathbf{e}_i\epsilon) = f(\mathbf{x}) + \epsilon\nabla_i f(\mathbf{x})$. Since computational algorithms can be interpreted as the composition of simpler functions, one need only define automatic differentiation on a small set of base cases (such as $+$, $*$, sin, and so forth, known as the primitives) and then apply the accepted rules in sequence to differentiate more elaborate functions. The ForwardDiff.jl package [20] in Julia accomplishes this by defining dispatches for such primitives on a dual number type and provides convenience functions for easily extracting common objects like gradients, Jacobians, and Hessians. Hessians are calculated by layering automatic differentiation twice on the same algorithm to effectively take the derivative of a derivative.

In this form, forward mode automatic differentiation shares many similarities to the complex perturbation methods described above without the requirement that the extension of $f(\mathbf{x})$ be complex analytic. At every stage of the calculation $f(\mathbf{x})$ must be differentiable, a weaker yet still restrictive assumption. Conveniently, automatic differentiation allows for arbitrarily many derivatives to be calculated simultaneously. By defining higher-dimensional dual numbers that act independently via

$$f(a + \sum_i b_i\,\epsilon_i) = f(a) + \sum \epsilon_i df(a)b_i$$

one can calculate entire Jacobians in a single function call $f(a + \sum_i e_i\,\epsilon_i)$. This use of higher-

dimensional dual numbers is a practice known as chunking. Chunking reduces the number of primal (non-derivative) calculations required for computing the Jacobian. Because the ForwardDiff.jl package uses chunking by default, we will investigate the extent to which this detail is applicable in biological models.

## 3 Case studies

We now explore applications of differential sensitivity to a few core models in oncology and epidemiology.

### 3.1 CARRGO model

The CARRGO model [21] was designed to capture the tumor-immune dynamics of CAR T-cell therapy in glioma. The CARRGO model generalizes to other tumor cell-immune cell interactions. Its governing system of ODEs

$$
\begin{aligned}
\frac{dx}{dt} &= \rho x \left( 1 - \frac{y}{\gamma} \right) - \kappa_1 xy \\
\frac{dy}{dt} &= \kappa_2 xy - \theta y
\end{aligned}
$$

follows cancer cells $x$ as prey and CAR T-Cells $y$ as predators. This model captures Lotka-Volterra dynamics with logistic growth of the cancer cells. Our numerical experiments assume the parameter values and initial conditions

$$
\begin{aligned}
\kappa_1 &= 6 \times 10^{-9}/(\text{day} \times \text{cell}), \quad \kappa_2 = 3 \times 10^{-11}/(\text{day} \times \text{cell}), \\
\theta &= 1 \times 10^{-6}/\text{day}, \quad \rho = 6 \times 10^{-2}/\text{day}, \quad \gamma = 1 \times 10^{9} \text{cells}, \\
x_0 &= 1.25 \times 10^{4} \text{cells}, \quad y_0 = 6.25 \times 10^{2} \text{cells}
\end{aligned}
$$

suggested by Sahoo et al. [21].

A traditional sensitivity analysis hinges on solving the system of ODEs and displaying the solutions at a chosen future time across an interval or rectangle of parameter values. Fig 1 shows how $x(t)$ and $y(t)$ vary at $t = 1000$ days under joint changes of $\kappa_1$ and $\kappa_2$, where $\kappa_1$ is the rate at which cancer cells are destroyed in an interaction with an immune cell, and $\kappa_2$ is the rate at which immune cells are recruited after such an interaction. This type of analysis directly portrays how a change in one or two parameters impacts the outcome of the system. Surprisingly, the number of cancer cells $x(t)$ depends strongly on $\kappa_2$ but only weakly on $\kappa_1$. In contrast, the number of immune cells $y(t)$ depends comparably on both parameters, perhaps because the initial population of immune cells is much smaller than the initial population of cancer cells.

There are limitations to this type of sensitivity analysis. How many solution curves should be examined? What time is most informative in displaying system changes? Is it necessary to compute sensitivity over such a large range of parameters when the trends are so clear? These ambiguities cloud our understanding and require far more computing than is necessary. Differential sensitivity successfully addresses these concerns. Gradients of solutions immediately yield approximate solutions in a neighborhood of postulated parameter values. The relative importance of different parameters in determining species levels can be determined from inspection of the gradient. Furthermore, modern software easily delivers the gradient along entire solution trajectories. There is no need to solve for an entire bundle of neighboring solutions.

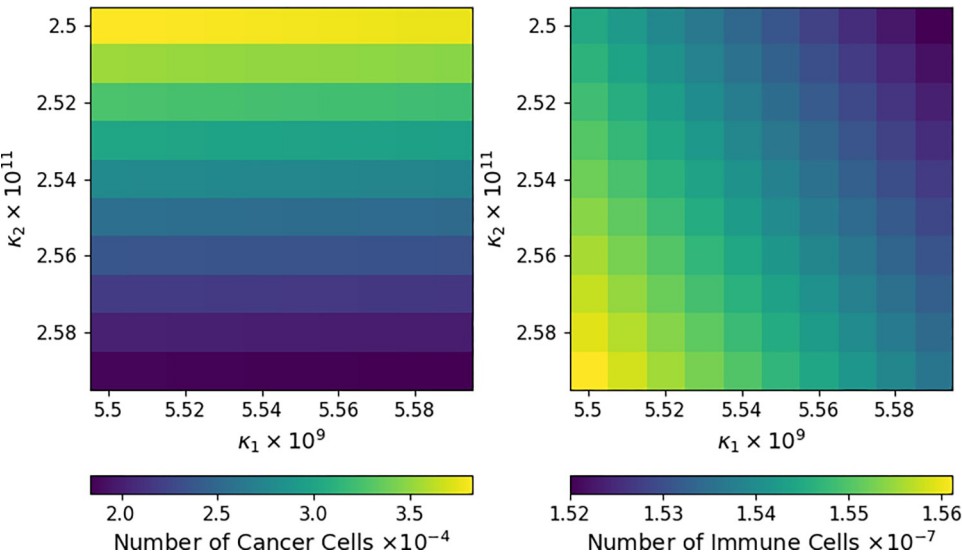

**Fig 1. Sensitivity of Cancer and Immune Cells in the CARRGO Model.** A heatmap representing the number of cancer cells, or $x(t)$ (left) and the number of immune cells, or $y(t)$ (right) as the parameters $\kappa_1$ (horizontal axis) and $\kappa_2$ (vertical axis) are varied. Results displayed summarize simulations of the CARRGO model with parameter values and initial conditions indicated in this section at time $t = 1000$ days.

Differential assessment is far more efficient. The required calculations involve solving an expanded system of ordinary differential equations just once under the automatic differentiation method or solving the system once for each parameter under the complex perturbation method. Either way, the differential method is much less computationally intensive than the traditional method of solving the ODE system over an interval for each parameter or over a rectangle for each pair of parameters. Here is our brief Julia code for computing sensitivity via the complex perturbation method.

```julia
using DifferentialEquations, Plots
function sensitivity(x0, p, d, tspan)
    problem = ODEProblem{true}(ODE, x0, tspan, p)
    sol = solve(problem, saveat = 1.0) # solve ODE
    (lp, ls, lx) = (length(p), length(sol), length(x0))
    solution = Dict{Int, Any}(i = > zeros(ls, lp + 1) for i in 1:lx)
    for j = 1:lx # record solution for each species
        @views solution[j][:, 1] = sol[j,:]
    end
    for j = 1:lp
        p[j] = p[j] + d * im # perturb parameter
        problem = ODEProblem{true}(ODE, x0, tspan, p)
        sol = solve(problem, saveat = 1.0) # resolve ODE
        p[j] = complex(real(p[j]), 0.0) # reset parameter
        @views sol. = imag(sol) / d # compute partial
        for k = 1:lx # record partial for each species
            @views solution[k][:,j + 1] = sol[k,:]
        end
    end
    return solution
end
function ODE(dx, x, p, t) # CARRGO model
    dx[1] = p[4] * x[1] * (1−x[1] / p[5])−p[1] * x[1] * x[2]
    dx[2] = p[2]* x[1] * x[2]−p[3] * x[2]
end
```

```
p = complex([6.0e-9, 3.0e-11, 1.0e-6, 6.0e-2, 1.0e9]); # parameters
x0 = complex([1.25e4, 6.25e2]); # initial values
(d, tspan) = (1.0e-13, (0.0,1000.0)); # step size and time interval in
days
solution = sensitivity(x0, p, d, tspan); # find solution and partials
CARRGO1 = plot(solution[1][:, 1], label = "x1", xlabel = "days",
ylabel = "cancer cells x1", xlims = (tspan[1],tspan[2]))
CARRGO2 = plot(solution[1][:, 2], label = "d1x1", xlabel = "days",
ylabel = "p1 sensitivity", xlims = (tspan[1],tspan[2]))
```

In the Julia code the parameters $\kappa_1$, $\kappa_2$, $\theta$, $\rho$, and $\gamma$ and the variables $x$ and $y$ exist as components of the vector **p** and **x**, respectively. The two plot commands construct solution curves for cancer and its sensitivity to perturbations of $\kappa_1$.

Fig 2 reinforces the conclusions drawn from the heatmaps, but more clearly and quantitatively. Additionally, differential sensitivity allows for the assessment of the sensitivity over the course of time, rather than just at a single time or small set of times. For example, the sensitivity of $x$ with respect to $\gamma$ in this model exhibits both large positive and large negative values over the course of time. Measured as the difference in $x$ caused by a difference in $\gamma$ at our end-time $t = 1000$, these effects tend to cancel each other out and fail to communicate the impact of the parameter $\gamma$ on $x$ at intermediate times. In brief, the scaled sensitivity of cancer cells $x$ is much more dependent on carrying capacity $\gamma$ later in the simulation, while the model sensitivity to birth rate $\rho$ is most pronounced around the earlier time $t = 200$.

### 3.2 Deterministic SIR model

The deterministic SIR model follows the number of infectives $I(t)$, the number of susceptibles $S(t)$, and the number of recovereds $R(t)$ during an epidemic. These three subpopulations satisfy the ODE system

$$
\begin{aligned}
\frac{d}{dt}S &= -\eta I \frac{S}{N} \\
\frac{d}{dt}I &= \eta I \frac{S}{N} - \delta I \\
\frac{d}{dt}R &= \delta I,
\end{aligned}
$$

where $\eta$ is the daily infection rate per encounter and $\delta$ is the daily rate of progression to immunity per person. For SARS-CoV-2, current estimates [22] of $\eta$ range from 0.0012 to 0.48, and estimates of $\delta$ range from 0.0417 to 0.0588 [23]. As an alternative to solving the extended set of differential equations, we again use the complex perturbation method to evaluate parameter sensitivities.

The following Julia code for the complex perturbation method reuses the generic sensitivity function from the CARRGO model example.

```
function ODE(dx, x, p, t) # Covid model
    N = 3.4e8 # US population size
    dx[1] = −p[1] * x[2] * x[1] / N
    dx[2] = p[1] * x[2] * x[1] / N−p[2] * x[2]
    dx[3] = p[2] * x[2]
end
p = complex([0.2, (0.0417 + 0.0588) / 2]); # parameters
x0 = complex([3.4e8, 100.0, 0.0]); # initial values
(d, tspan) = (1.0e-10, (0.0, 365.0)) # 365 days
solution = sensitivity(x0, p, d, tspan);
Covid = plot(solution[1][:,:], label = ["x1" "d1x1" "d2x1"],
xlabel = "days", xlims = (tspan[1],tspan[2]))
```

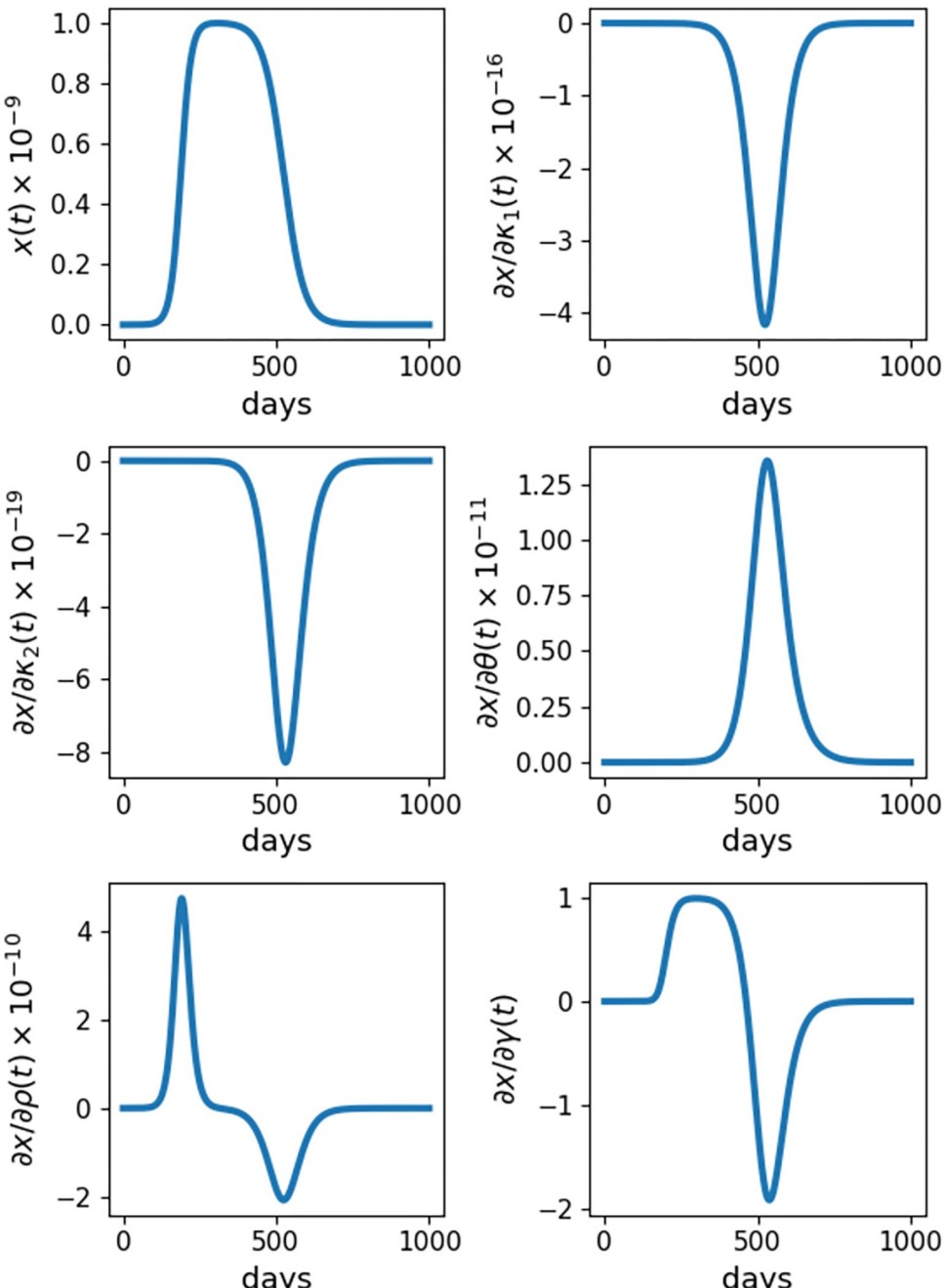

**Fig 2. Sensitivity of Cancer Cells in the CARRGO Model.** Time series plots of cancer cells ($x(t)$) and the derivatives of $x(t)$ with respect to the CARRGO parameters $\kappa_1$, $\kappa_2$, $\theta$, $\rho$, $\gamma$. Results shown are for the initial conditions and parameter values defined in Fig 1 and simulated over the course of $t = 1000$ days. The complex perturbation method of sensitivity analysis is used to compute derivatives.

Our parameter choices roughly capture measurements for the COVID-19 virus from early in the pandemic [22,23]. Fig 3 plots the susceptible curve and its sensitivities. In this case all three curves conveniently occur on comparable scales. Fig 3 captures not only the pronounced parameter sensitivity early in the pandemic but also the symmetry between the $\delta$ and $\eta$ parameters.

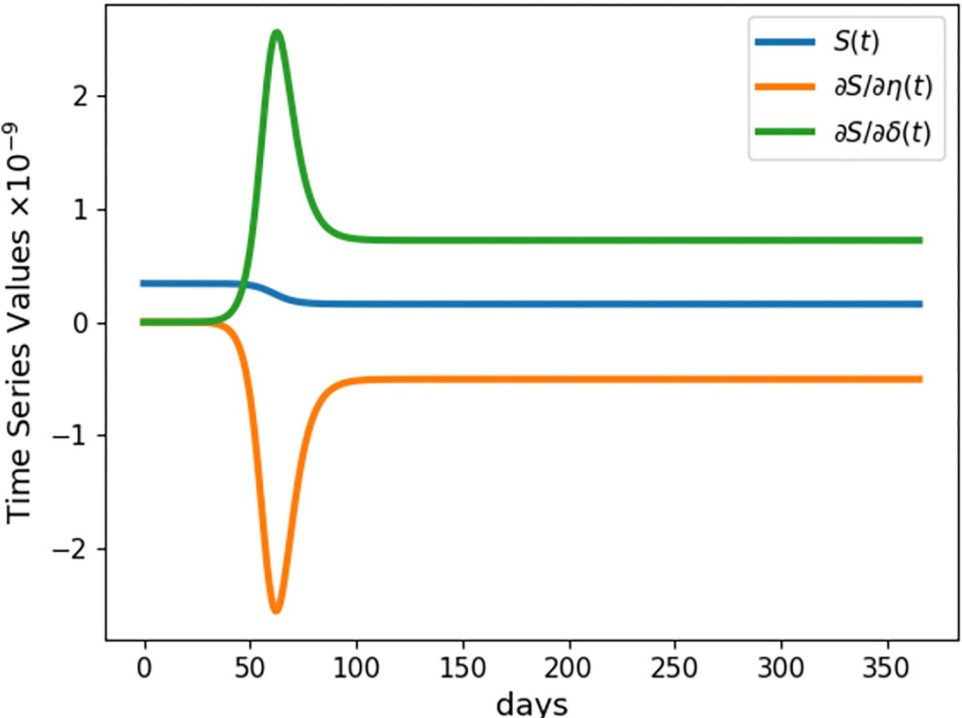

**Fig 3. Sensitivities of Susceptibles in the Covid Model.** Time series of the susceptible population ($S(t)$) and its sensitivities to the two parameters ($\eta$ and $\delta$) of the classic SIR model. Results shown are for the SIR model simulated for one year with initial conditions $S_0 = 3.4 \times 10^8$, $I_0 = 100$, $R_0 = 0$, and the parameter values $\eta = 0.7194$, $\delta = 0.5025$. Derivatives are calculated using the complex perturbation method.

### 3.3 Second-order expansions of solution trajectories

In predicting nearby solution trajectories, the second-order Taylor expansion

$$f(\mathbf{x} + \mathbf{v}) \approx f(\mathbf{x}) + df(\mathbf{x})\mathbf{v} + \frac{1}{2}\mathbf{v}^t d^2 f(\mathbf{x})\mathbf{v} \qquad (4)$$

improves accuracy over the first-order expansion

$$f(\mathbf{x} + \mathbf{v}) \approx f(\mathbf{x}) + df(\mathbf{x})\mathbf{v}. \qquad (5)$$

The improved accuracy achieved by including second-order terms often justifies their computation. The complex perturbation method permits straightforward computation of second derivatives via approximations (Eq 2) and (Eq 3). The DiffEqSensitivity.jl and ForwardDiff.jl packages implement both adjoint and forward difference methods for computing the second derivatives of differential equation systems. Fig 4 displays predicted trajectories for the SIR model using the complex perturbation method when all parameters $p_i$ are replaced by $p_i(1 + U_i)$, where each $U_i$ is chosen uniformly from ($-0.25, 0.25$). Fig 4 vividly confirms the improvement in accuracy in passing from a first-order to a second-order approximation. More improvement becomes evident as the non-linearity of the solution trajectory increases.

For example, the top right panel of Fig 4 shows that the solution trajectory of infected individuals bends dramatically with a change in parameters. This behavior is much better reflected in the second-order prediction compared to the first-order prediction, which over-corrects at the peak. The Euclidean distance between the actual and predicted trajectories at the sampled time points is about 25.4 in the first-order case and only about 9.06 in the second-order case, a

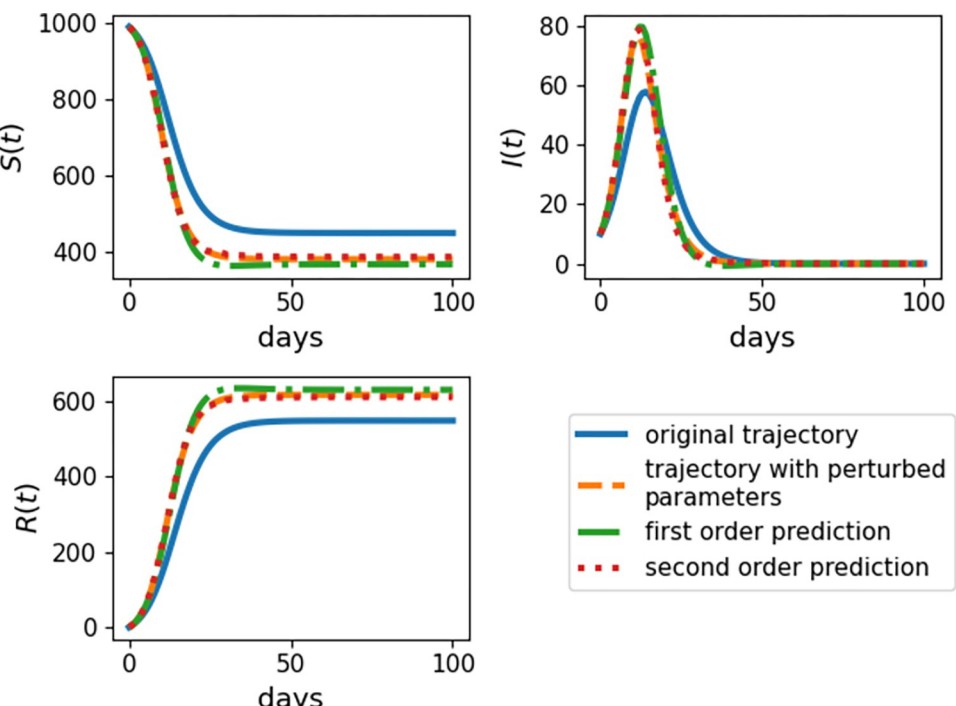

**Fig 4. Model Trajectories for SIR Model Calculated Using First and Second Differentials.** Time series plot of the SIR model simulated over $t = 100$ days with initial conditions $S_0 = 1000$ and $I_0 = 10$. Results depend on the SIR model with the original parameters from Fig 3 (original trajectory), re-simulating the SIR trajectory after perturbing the parameters by a random amount around 25% (trajectory with perturbed parameters), approximating the trajectory based on the linear expansion (Eq 5) and the first derivative calculated with the complex perturbation method (first-order prediction), and approximating the trajectory based on the quadratic expansion (Eq 4) and the first and second derivatives calculated with the complex perturbation method (second-order prediction).

reduction of over 60% in prediction error. By contrast, the trajectory of the recovered individuals steadily increases in a much more linear fashion. The bottom left panel of Fig 4 shows that the first-order prediction now remains reasonably accurate over a substantial period. Even so, the discrepancy between the predicted solutions grows so that by day 100 the Euclidean distance between the first-order prediction and the actual trajectory exceeds 154, compared to about 34.0 for the second-order prediction. Thus, calculating second-order sensitivity is helpful in both highly non-linear systems and systems with long time scales.

## 3.4 Stochastic SIR model

We now illustrate sensitivity calculations in the stochastic SIR model. This model postulates an original population of size $n$ with $i$ infectives and $s$ susceptibles. The parameters $\delta$ and $\eta$ again capture the rate of progression to immunity and the infection rate per encounter. Since extinction of the infectives is certain, we focus on the time to elimination of the infectives. It is also convenient to follow the vector $(i, n)$, where $n = i+s$ is the sum of the number of infectives $i$ plus the number of susceptibles $s$. The mean time $t_{in}$ to elimination of all infectives satisfies the recurrence

$$
t_{in} = \frac{1}{i\delta + i\left(\frac{n-i}{N}\right)\eta} + \frac{i\delta}{i\delta + i\left(\frac{n-i}{N}\right)\eta} t_{i-1,n-1}
$$
$$
+ \frac{i\left(\frac{n-i}{N}\right)\eta}{i\delta + i\left(\frac{n-i}{N}\right)\eta} t_{i+1,n} \tag{6}
$$

for $0<i<n$ together with the boundary conditions

$$t_{ii} \;=\; \sum_{j=1}^{i} \frac{1}{j\delta} \;\; \text{and} \;\; t_{0n} = 0.$$

The expression for $t_{ii}$ stems from adding the expected time for the $i \rightarrow i-1$ transition, plus the expected time $i-1 \rightarrow i-2$, and so forth. This system of equations can be solved recursively for $i = n, n-1, \ldots .0$ starting with $n = 1$. Once the values for a given $n$ are available, $n$ can be incremented, and a new round is initiated. Ultimately the target size $n = N$ is reached. Taking partial derivatives of the recurrence (Eq 6) yields a new system of recurrences that can also be solved recursively in tandem with the original recurrence. The complex perturbation method is easier to implement and comparable in accuracy to the partial derivative method.

Another important index of the SIR process is the mean number of infectives $m_{in}$ ever generated starting with $i$ initial infectives and $n$ total people. These expectations can be calculated via the recurrences

$$m_{in} \;=\; \frac{i\delta}{i\delta + i\left(\frac{n-i}{N}\right)\eta}\left(m_{i-1,n-1} + 1\right) + \frac{i\left(\frac{n-i}{N}\right)\eta}{i\delta + i\left(\frac{n-i}{N}\right)\eta} m_{i+1,n} \tag{7}$$

for $0<i<n$ together with the boundary conditions

$$m_{ii} \;=\; i \;\; \text{and} \;\; m_{0n} = 0.$$

One can compute the sensitivities of the $m_{in}$ to parameter perturbations in the same way as the $t_{in}$. Here is the Julia code for the two means and their sensitivities via the complex perturbation method. Note how our earlier differential function plays a key role.

```julia
function SIRMeans(p)
    (delta, eta) = (p[1], p[2])
    M = zeros(typeof(p[1]),(N+1, N+1)) # mean matrix
    T = similar(M) # time to extinction matrix
    for n = 1:N # recurrence relations loop
        for j = 0:(n-1)
            i = n−j
            a = i * delta # immunity rate
            if i == n # initial conditions
                M[i+1, n+1] = i
                T[i+1, n+1] = T[i, i] + 1 / a
            else
                b = i * (n−i) * eta / N # infection rate
                c = 1 / (a + b)
                M[i+1, n+1] = a * c * (M[i, n] + 1) + b * c * M[i+2, n+1]
                T[i+1, n+1] = c * (1 + a * T[i, n] + b * T[i+2, n+1])
            end
        end
    end
    return [M[:, N+1]; T[:, N+1]]
end
p = complex([0.2, (0.0417 + 0.0588) / 2]); # delta and beta
(N, d) = (100, 1.0e-10);
@time (f, df) = differential(SIRMeans, p, d);
```

The left column of Fig 5 displays a heatmap of the expected total number of individuals infected and the right column displays a heatmap of the expected days to extinction of the infection process. Rows 2 and 3 show the sensitivites of these quantities to the $\eta$ and $\delta$ parameters in the stochastic SIR model.

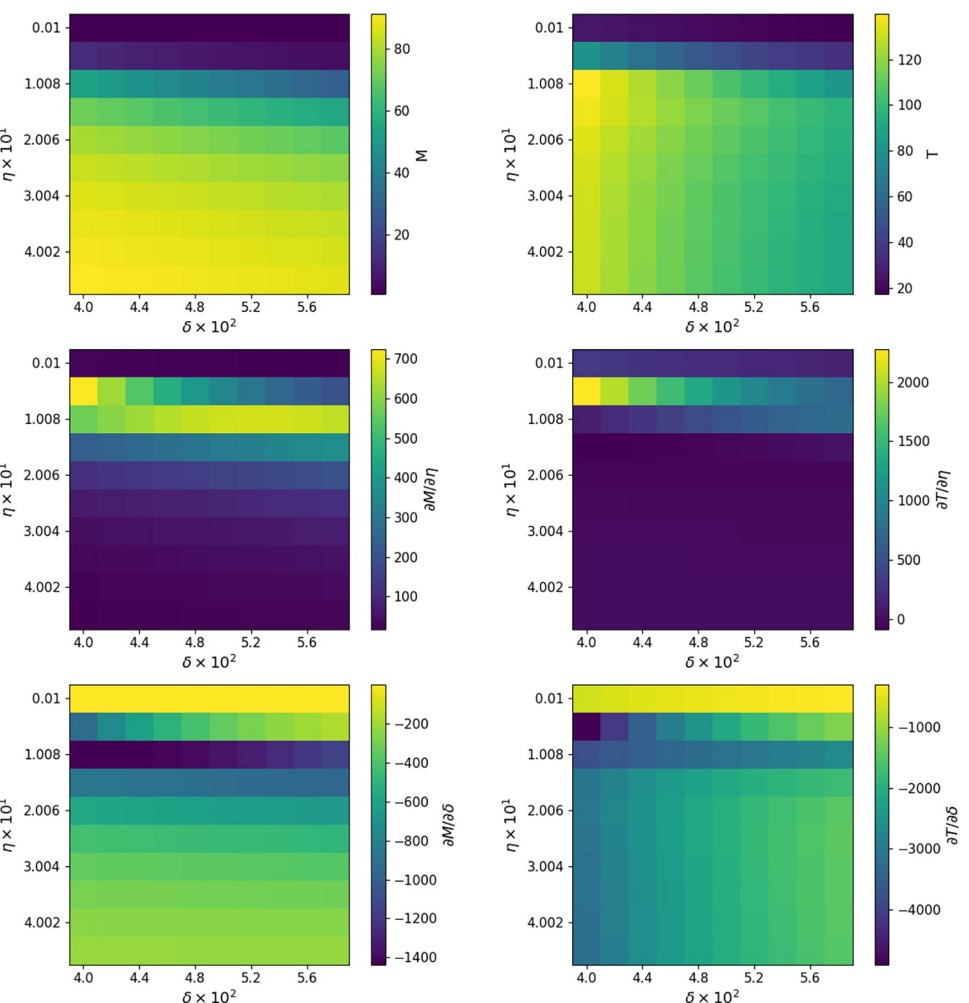

**Fig 5. Sensitivity of Stochastic SIR Model.** Heatmaps showing the mean number of infected individuals ($M$) at extinction, the mean time to extinction ($T$), and their sensitivities to the parameters $\eta$ and $\delta$ for the stochastic SIR process. Sensitivities rely on the complex perturbation method to calculate derivatives and assume initial conditions $S_0$ = 100 and $I_0$ = 1.

It is interesting to compare results from differential sensitivity to estimates from stochastic simulations. To see the difference in accuracy, we calculated the average number of individuals infected and the average time to extinction by stochastic simulation using the software package BioSimulator.jl [24]. Table 1 records the analytic and simulated means of these outcomes in the SIR model. As Table 1 indicates, the simulated means over $r$ = 100 runs are roughly comparable to the analytic means, but the standard errors of the simulated means are large. Because

**Table 1. Comparison between the calculated and simulated means of SIR model outcomes in the stochastic SIR model simulated under the initial conditions $S_0$ = $3.4 \times 10^4$, $I_0$ = 1 and parameter values $\eta$ = 0.7194, $\delta$ = .5025. Results for the simulated means were obtained using the BioSimulator package in Julia and averaging results over $r$ = 100 runs.**

|  | Calculated Mean | Simulated Mean | Simulated Standard Error |
| --- | --- | --- | --- |
| Time to Extinction | $2.792 \times 10$ days | $3.074 \times 10$ days | 4.153 days |
| Number Infected | $5.484 \times 10^3$ people | $5.838 \times 10^3$ people | $8.551 \times 10^2$ people |

the standard errors decrease as $\frac{1}{\sqrt{r}}$, it is difficult to achieve much accuracy by simulation alone. In more complicated models, simulation is so computationally intensive and time consuming that it is nearly impossible to achieve accurate results. Of course, the analytic method is predicated on the existence of an exact solution or an algorithm for computing the same.

Parameter sensitivities inform our judgment in interesting and helpful ways. For example, derivatives of both the total number of infecteds and the time to extinction with respect to $\eta$ are very small except in a narrow window of the $\eta$ parameter. This suggests that we focus further simulations, sensitivity analysis, and possible interventions on the region of parameter space where $\eta$ falls in these windows. Derivatives with respect to $\delta$ also depend mostly on $\eta$ except at very small values of $\delta$. These conclusions are harder to draw from noisy simulations alone.

### 3.5 Branching processes

Branching process models offer another opportunity for checking the accuracy of sensitivity calculations. For simplicity we focus on birth-death-migration processes [25]. These are multi-type continuous-time processes [26,17] that can be used to model the early stages of an epidemic over a finite graph with $n$ nodes, where nodes represent cities or countries. On node $i$ we initiate a branching process with birth rate $\beta_i > 0$ and death rate $\delta_i > 0$. The migration rate from node $i$ to node $j$ is $\lambda_{ij} \geq 0$. All rates are per person, and each person is labeled by a node. Let $\lambda_i = \sum_{j \neq i} \lambda_{ij}$ be the sum of the migration rates emanating from node $i$. Given this notation, the mean infinitesimal generator of the process is the matrix

$$\mathbf{\Omega} \;=\; \begin{pmatrix} \beta_1 - \delta_1 - \lambda_1 & \lambda_{12} & \cdots & \lambda_{1,n-1} & \lambda_{1n} \\ \vdots & \vdots & \ddots & \vdots & \vdots \\ \lambda_{n1} & \lambda_{n2} & \cdots & \lambda_{n,n-1} & \beta_n - \delta_n - \lambda_n \end{pmatrix}$$

The entries of the matrix $e^{t\mathbf{\Omega}} = [m_{ij}(t)]$ represent the expected number of people at node $j$ at time $t$ starting from a single person of type $i$ at time 0. The process is irreducible when the pure migration process corresponding to the choice $\beta_i = \delta_i = 0$ for all $i$ is irreducible. Equivalently, the process is irreducible when the graph representing transition probabilities is strongly connected. Henceforth, we assume the process is irreducible and let $\mathbf{\Gamma}$ denote the mean infinitesimal generator of the pure migration process. The process is subcritical, critical, or supercritical depending on whether the dominant eigenvalue $\rho$ of $\mathbf{\Omega}$ is negative, zero, or positive.

To determine the local sensitivity of $\rho$ to a parameter $\theta$ [26, 27], suppose its left and right eigenvectors $\mathbf{v}$ and $\mathbf{w}$ are normalized so that $\mathbf{v}\mathbf{w} = 1$. Differentiating the identity $\mathbf{\Omega}\mathbf{w} = \rho\mathbf{w}$ with respect to $\theta$ yields

$$\left(\frac{\partial}{\partial\theta}\mathbf{\Omega}\right)\mathbf{w} + \mathbf{\Omega}\frac{\partial}{\partial\theta}\mathbf{w} \;=\; \left(\frac{\partial}{\partial\theta}\rho\right)\mathbf{w} + \rho\frac{\partial}{\partial\theta}\mathbf{w}.$$

If we multiply this by $\mathbf{v}$ on the left and invoke the identities $\mathbf{v}\mathbf{\Omega} = \rho\mathbf{v}$ and $\mathbf{v}\mathbf{w} = 1$ we find that

$$\frac{\partial}{\partial\theta}\rho \;=\; \mathbf{v}\left(\frac{\partial}{\partial\theta}\mathbf{\Omega}\right)\mathbf{w}.$$

Because $\frac{\partial}{\partial\delta_i}\mathbf{\Omega} = -\frac{\partial}{\partial\beta_i}\mathbf{\Omega}$, it follows that an increase in $\delta_i$ has the same impact on $\rho$ as the same decrease in $\beta_i$. The sensitivity of $\mathbf{v}$ and $\mathbf{w}$ can be determined by an extension of this reasoning [28]. The extinction probabilities $e_i$ of the birth-death-migration satisfy the system of

algebraic equations

$$e_i = \frac{\delta_i}{\beta_i + \delta_i + \lambda_i} + \frac{\beta_i}{\beta_i + \delta_i + \lambda_i} e_i^2 + \sum_{j \neq i} \frac{\lambda_{ij}}{\beta_i + \delta_i + \lambda_i} e_j \qquad (8)$$

for all $i$. This is a special case of the vector extinction equation

$$\mathbf{e} = P(\mathbf{e}) = \begin{pmatrix} P_1(\mathbf{e}) \\ \vdots \\ P_n(\mathbf{e}) \end{pmatrix}$$

for a general branching process with offspring generating function $P_i(\mathbf{x})$ for a type $i$ person [29]. For a subcritical or critical process, $\mathbf{e} = \mathbf{1}$. For a supercritical process all $e_i \in (0,1)$. Iteration is the simplest way to find $\mathbf{e}$. Starting from $\mathbf{e}_0 = \mathbf{0}$, the vector sequence $\mathbf{e}_n = P(\mathbf{e}_{n-1})$ satisfies

$$0 \leq \mathbf{e}_n \leq \mathbf{e}_{n+1} \leq \mathbf{e}$$

and converges to a solution of the extinction equations. Here all inequalities apply component-wise.

To find the differential [28] of the extinction vector $\mathbf{e}$ with respect to a vector $\boldsymbol{\theta}$ of parameters, we assume that the branching process is supercritical and resort to implicit differentiation of the equation $\mathbf{e}(\boldsymbol{\theta}) = P[\mathbf{e}(\boldsymbol{\theta}), \boldsymbol{\theta}]$. The chain rule gives

$$d_{\boldsymbol{\theta}}\mathbf{e} = d_{\mathbf{e}}P(\mathbf{e}, \boldsymbol{\theta})d_{\boldsymbol{\theta}}\mathbf{e} + d_{\boldsymbol{\theta}}P(\mathbf{e}, \boldsymbol{\theta}).$$

This equation has the solution

$$d_{\boldsymbol{\theta}}\mathbf{e} = [\mathbf{I}_n - d_{\mathbf{e}}P(\mathbf{e}, \boldsymbol{\theta})]^{-1} d_{\boldsymbol{\theta}}P(\mathbf{e}, \boldsymbol{\theta}). \qquad (9)$$

The indicated inverse does, in fact, exist. Alternatively, one can compute an entire extinction curve $\mathbf{e}(t)$ whose component $e_i(t)$ supplies the probability of extinction before time $t$ starting from a single person of type. This task reduces to solving the ODE for $\frac{d}{dt}\mathbf{e}(t)$ by the methods previously discussed.

The following Julia code computes the sensitivities of the extinction probability for a two-node process by the complex perturbation method.

```
using LinearAlgebra
function extinction(p)
    types = Int(sqrt(1 + length(p)) - 1) # length(p) = 2 * types + types^2
    (x, y) = (zeros(Complex, types), zeros(Complex, types))
    for i = 1:500 # functional iteration
        y = P(x, p)
        if norm(x−y) < 1.0e-16 break end
            x = copy(y)
        end
        return y
    end
function P(x, p) # progeny generating function
    types = Int(sqrt(1 + length(p)) - 1) # length(p) = 2 * types
+ types^2
    delta = p[1: types]
    beta = p[types + 1: 2 * types]
    lambda = reshape(p[2 * types + 1:end], (types, types))
    y = similar(x)
```

```
        t = delta[1] + beta[1] + lambda[1, 2]
        y[1] = (delta[1] + beta[1] * x[1]^2 + lambda[1, 2] * x[2]) / t
        t = delta[2] + beta[2] + lambda[2, 1]
        y[2] = (delta[2] + beta[2] * x[2]^2 + lambda[2, 1] * x[1]) / t
        return y
    end
    delta = complex([1.0, 1.75]); # death rates
    beta = complex([1.5, 1.5]); # birth rates
    lambda = complex([0.0 0.5; 1.0 0.0]); # migration rates
    p = [delta; beta; vec(lambda)]; # package parameter vector
    (types, d) = (2, 1.0e-10)
    @time (e, de) = differential(extinction, p, d)
```

To adapt the code to a different branching process model, one simply supplies the appropriate progeny generating function and necessary parameters.

The average number $a_{ij}$ of infected individuals of type $j$ ultimately generated by a single initial infected individual of type $i$ is also of interest. The matrix $\mathbf{A} = (a_{ij})$ of these expectations can be calculated via the matrix equation

$$\mathbf{A} \;=\; (\mathbf{I}_n - \mathbf{F})^{-1}, \tag{10}$$

where $\mathbf{F}$ is the offspring matrix

$$\mathbf{F} \;=\; \begin{pmatrix} \dfrac{2\beta_1}{\beta_1 + \delta_1 + \lambda_1} & \dfrac{\lambda_{12}}{\beta_1 + \delta_1 + \lambda_1} & \cdots & \dfrac{\lambda_{1,n-1}}{\beta_1 + \delta_1 + \lambda_1} & \dfrac{\lambda_{1n}}{\beta_1 + \delta_1 + \lambda_1} \\ \vdots & \vdots & \ddots & \vdots & \vdots \\ \dfrac{\lambda_{n1}}{\beta_n + \delta_n + \lambda_n} & \dfrac{\lambda_{n2}}{\beta_n + \delta_n + \lambda_n} & \cdots & \dfrac{\lambda_{n,n-1}}{\beta_n + \delta_n + \lambda_n} & \dfrac{2\beta_n}{\beta_n + \delta_n + \lambda_n} \end{pmatrix}.$$

One can determine the local sensitivity of the expected numbers of total descendants by differentiating the equation $\mathbf{A} = (\mathbf{I}_n - \mathbf{F})^{-1}$. The result

$$d_{\boldsymbol{\theta}}\mathbf{A} \;=\; (\mathbf{I}_n - \mathbf{F})^{-1} d_{\boldsymbol{\theta}}\mathbf{F}(\mathbf{I} - \mathbf{F})^{-1}, \tag{11}$$

depends on the sensitivity of the expected offspring matrix $\mathbf{F}$. Julia code for the complex perturbation method with two nodes follows.

```
function particles(p) # mean infected individuals generated
    types = Int(sqrt(1 + length(p)) - 1) # length(p) = 2 * types + types^2
    delta = p[1: types]
    beta = p[types + 1: 2 * types]
    lambda = reshape(p[2 * types + 1:end], (types, types))
    F = complex(zeros(types, types))
    t = delta[1] + beta[1] + lambda[1, 2]
    (F[1, 1], F[1, 2]) = (2 * beta[1] / t, lambda[1, 2] / t)
    t = delta[2] + beta[2] + lambda[2, 1]
    (F[2, 1], F[2, 2]) = (lambda[2, 1] / t, 2 * beta[2] / t)
    A = vec(inv(I−F)) # return as vector
end
delta = complex([1.0, 1.75]); # death rates
beta = complex([1.5, 1.5]); # birth rates
lambda = complex([0.0 0.5; 1.0 0.0]); # migration rates
p = [delta; beta; vec(lambda)]; # package parameter vector
(types, d) = (2, 1.0e-10)
@time (A, dA) = differential(particles, p, d)
```

## 4 Results

We now measure the accuracy, computational speed, and prediction error for adjoint, forward mode, and complex perturbation methods. To account for the variety of settings encountered by biologists, we include two additional ODE models in our comparisons. The ROBER model describes chemical reactions typical of enzymatic behavior [30] and furnishes an example of a stiff ODE system. More information on the ROBER model can be found in S2 Appendix. To compare the three methods in a high-dimensional ODE model, we turn to the mammalian cell cycle (MCC) model. Our MCC model is a simplified version of the original MCC model constructed by Gerard and Goldbetor [31], as explained in more detail in S2 Appendix. The model comprises 11 equations and 15 parameters and captures aspects of cell reproduction and cycling mediated by chemical signaling via cell-state dependent proteins such as tumor repressors, transcription factors, and other DNA replication checkpoints. The model relies on cell state as opposed to cell mass and nicely replicates sequential progression along the cell cycle.

### 4.1 Accuracy

It is important to understand how close computed differential sensitivities are to true differential sensitivities. Unfortunately, the latter are almost always unavailable for ODE models. For the stochastic SIR and branching process models, true sensitivities are well matched by the approximate sensitivities delivered by the complex perturbation methods, provided the complex perturbation is small enough [32]. As a proxy for comparison to true values in ODE models, one can compute the Euclidean distance between sensitivities delivered by the complex perturbation method and the methods relying on the chain rule. In general, we find that these distances are very small.

For the forward and adjoint sensitivities of non-stiff ODEs such as the SIR and CARRGO models, it is known that as one decreases the tolerance of the underlying ODE solver, the solution and its sensitivities converge to their true values [33]. To demonstrate that the same behavior occurs in our cases, we compute the sensitivities $\frac{\partial}{\partial \eta} S$ of the SIR model and $\frac{\partial}{\partial p_1} x_1$ of the ROBER model at $t = 1000$ using the adjoint, forward, and complex perturbation methods at a variety of tolerances ranging from $1 \times 10^{-2}$ to $1 \times 10^{-8}$.

Fig 6 shows that all three method types (adjoint, forward, and complex perturbation) ultimately converge. In the non-stiff case (the SIR model), the adjoint method requires a step size of 1.0 to converge, while the stiff case (the ROBER model) requires a much smaller step size of 0.1 to converge. Each method converges at a different rate and potentially from a different direction. In the case of a relatively small, non-stiff model, the complex perturbation method converges more quickly (and at a higher tolerance) than the other methods. Notably, when the tolerance for the adjoint method is too weak the error rate increases more dramatically than for the forward method. This behavior becomes even more pronounced if we consider a stiff ODE model such as ROBER. In this case it is worth noting that the forward and complex perturbation methods converge, albeit under a more stringent tolerance. The adjoint method however struggles to converge for the ROBER model unless the step size is decreased to 0.1 (as shown in the Fig 6). While the smaller step size does allow the adjoint method to converge even in the stiff case, this smaller step size is much more computationally intensive and, in many cases, may be infeasible.

### 4.2 The speed versus accuracy trade-off

The trade-off between computational speed and accuracy is relevant to solving ODE systems whether they are stiff or not. Fig 7 displays the time versus error trade-off for both the SIR (non-stiff) and ROBER (stiff) models. In this case, error is calculated as the Euclidean distance

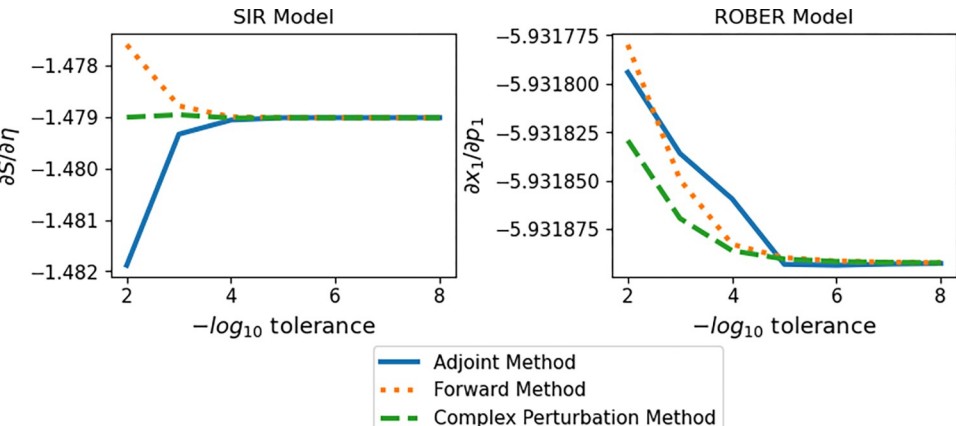

**Fig 6. Convergence of Adjoint, Forward, and Complex Perturbation Methods for Numerical Sensitivities.**
Convergence plot of the SIR model (left) and ROBER model (right) simulated over $t = 1000$ days. For SIR the initial conditions are $S_0 = 3.4\times10^8$, and $I_0 = 100$, and the parameters are $\eta = 0.7194$ and $\delta = 0.5025$. For ROBER the initial conditions are $x_1 = 1.0$, $x_2 = 0.0$, and $x_3 = 0.0$, and the parameters are $p_1 = 4\times10^{-2}$, $p_2 = 3\times10^7$, and $p_3 = 1\times10^4$. First-order sensitivities are computed via code from this manuscript (complex perturbation method), the ForwardDiff.jl package (forward method), and the Rodas4(autodiff = false) solver under the QuadratureAdjoint (autojacvec = EnzymeVJP()) sensealg protocol in the DiffEqSensitivities.jl package (adjoint method). The adjoint method requires a step size of 1.0 for the SIR model and a step size of 0.1 in the ROBER model to converge. All results are normalized by the number of time steps included in the simulation.

between the derivatives calculated at various error tolerances and the derivatives calculated at a strict tolerance of $1\times10^{-8}$ (for the SIR model) and $1\times10^{-5}$ (for the ROBER model). We chose these tolerances as the strictest possible that are numerically realistic for each model. Fig 6 demonstrates that our choices are strict enough for the methods to reach convergence. We display errors versus time in a log-log plot averaged over compartments and parameters and normalized by length of time. We do not include the adjoint method in this comparison due to its difficulties in convergence and large computational cost.

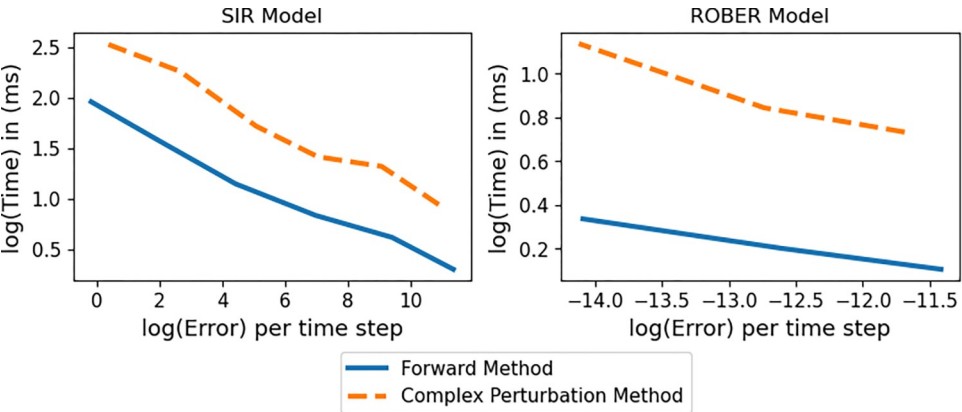

**Fig 7. Time vs Error of Forward and Complex Perturbation Methods for Numerical Sensitivities.** Time versus error log-log plot of the SIR model (left) and ROBER model (right) simulated over $t = 1000$ days. For SIR the initial conditions are $S_0 = 3.4\times10^8$, and $I_0 = 100$, and the parameters are $\eta = 0.7194$ and $\delta = 0.5025$. For ROBER the initial conditions are $x_1 = 1.0$, $x_2 = 0.0$, and $x_3 = 0.0$, and the parameters are $p_1 = 4\times10^{-2}$, $p_2 = 3\times10^7$, and $p_3 = 1\times10^4$. First-order sensitivities are computed via code from this manuscript (complex perturbation method) and the ForwardDiff.jl package (forward method). Times reported are the median times computed using the Benchmark.jl package, and errors are the Euclidean distance between the solution at the strictest tolerance ($10^{-8}$ for SIR and $10^{-5}$ for ROBER) and the solution at a variety of tolerances with a maximum of $10^{-2}$. All errors are normalized by the number of time steps.

Fig 7 demonstrates the clear trade-off between speed and accuracy in both the stiff (ROBER) and non-stiff (SIR) cases. In both cases, the forward method can be computed more quickly for equal errors than the complex perturbation method. As expected, the ROBER model has a less steep slope compared to the SIR model, indicating that the returns in accuracy grow more slowly per time invested for a stiff ODE system.

## 4.3 Computational speed

Speed is an important attribute of any computational method, especially when it is performed without the benefit of computational clusters or distributed computing resources. Our speed comparisons offer a first look at the efficiency gains possible with multithreading. In implementing multithreading for both the complex perturbation and forward mode methods, we call the Polyester.jl package to compute each partial derivative in a separate thread. All computations were done in Julia version 1.7.1 on a Windows operating system with an Intel Core i7-8565U CPU.

In addition to multithreading, the forward method as implemented in ForwardDiff.jl package provides the capability of multichunking. This involves splitting the equations in each system into different chunks to be solved separately. While forward methods do benefit from chunking, this tactic is unavailable in many packages outside of ForwardDiff.jl or outside of the Julia language. For biologists who depend on other packages and computer languages, it may be more pertinent to focus on the non-chunked results for the forward method.

Tables 2, 3, 4 and 5 record the computational speed of the complex perturbation, forward, and adjoint methods (and their multithreaded and multi-chunked versions, as applicable) for four ODE systems models (SIR, CARRGO, ROBER, and MCC). Our comparisons of the first-order methods show that the forward and complex perturbation methods perform comparably, while the adjoint method performs orders of magnitude slower than the other two. The fastest method is the multichunked forward method, with the complex perturbation method a close second for the simpler ODE systems such as SIR and CARRGO. For the stiff (ROBER) and large (MCC) models however, the complex perturbation method falls further behind the multichunk forward mode method. This could be expected from the larger gap between the time versus accuracy curves in the ROBER model as compared with the SIR model and illustrated in Fig 7. However, it is noteworthy that naive implementations of forward mode differentiation lack the advantage of chunking and are consequently slower than the complex perturbation method.

The adjoint method also has the worst time performance of the second-order methods by orders of magnitude. Both the forward and complex perturbation methods performed well in all four ODE systems models, with the complex perturbation method performing particularly well in models where the number of parameters is not large compared to the number of equations.

While multi-threading usually decreases computational time for both first-order and second-order methods, it does not decrease computational time by as wide of a margin as expected. Many of the solver methods for stiff ODEs rely on BLAS operations that are already internally optimized by running on multiple threads. Explicitly multi-threading sensitivity methods therefore restricts the number of threads available for BLAS operations, adversely affecting their performance. In addition to the reduced efficiency of BLAS operations, multi-threading incurs a start-up cost for each thread. These start-up costs may overshadow the benefits of multi-threading if the amount of computation per thread is not high enough. Multi-threaded methods require more allocations than other methods, and thus require more garbage collection. While time spent on garbage collection varies, we find that garbage collection

**Table 2. Computational time (μs) for the SIR ODE model.** Parameters match those previously introduced in this manuscript. Multithread refers to parallelism across parameters. Multichunk refers to parallelism across compartments. We invoke the Julia solver AutoVern9(Rodas5(autodiff = false)) with a tolerance of $1{\times}10^{-5}$, reflecting the convergence tolerance. For the second-order adjoint method, the ForwardDiffOverAdjoint(QuadratureAdjoint(autodiff = false) solver option was used.

| First-order Methods | $t_{end} = 10$ | $t_{end} = 100$ | $t_{end} = 1000$ |
|---|---|---|---|
| Complex Perturbation | $2.252 \times 10^2$ | $1.688 \times 10^3$ | $1.377 \times 10^4$ |
| Complex Perturbation Multithread | $1.913 \times 10^2$ | $1.401 \times 10^3$ | $1.062 \times 10^4$ |
| Forward | $3.272 \times 10^2$ | $2.036 \times 10^3$ | $1.460 \times 10^4$ |
| Forward Multithread | $2.218 \times 10^2$ | $1.480 \times 10^3$ | $1.117 \times 10^4$ |
| Forward Multichunk | $1.567 \times 10^2$ | $9.564 \times 10^2$ | $7.247 \times 10^3$ |
| Forward Multichunk Multithread | $1.499 \times 10^2$ | $9.526 \times 10^2$ | $7.236 \times 10^3$ |
| Adjoint | $8.901 \times 10^4$ | $7.707 \times 10^6$ | $6.950 \times 10^8$ |
| **Second-order Methods** | $t_{end} = 10$ | $t_{end} = 100$ | $t_{end} = 1000$ |
| Complex Perturbation | $7.885 \times 10^2$ | $5.712 \times 10^3$ | $5.806 \times 10^4$ |
| Complex Perturbation Multithread | $6.732 \times 10^2$ | $4.528 \times 10^3$ | $3.724 \times 10^4$ |
| Forward | $9.325 \times 10^2$ | $5.280 \times 10^3$ | $4.530 \times 10^4$ |
| Forward Multithread | $7.546 \times 10^2$ | $3.504 \times 10^3$ | $2.640 \times 10^4$ |
| Forward Multichunk | $1.742 \times 10^2$ | $7.601 \times 10^2$ | $4.541 \times 10^3$ |
| Forward Multichunk Multithread | $1.714 \times 10^2$ | $7.270 \times 10^2$ | $4.631 \times 10^3$ |
| Adjoint | $2.976 \times 10^4$ | $6.240 \times 10^5$ | $1.626 \times 10^7$ |

can take over twice as much computational time in multi-threaded methods than in their single-threaded counterparts. Thus, multi-threading can only really start to improve computational efficiency when these additional costs are small compared to the cost of each computation. Multi-threading may even be less efficient in some cases.

**Table 3. Computational time (μs) for the CARRGO ODE model.** Parameters match those previously introduced in this manuscript. Multithread refers to parallelism across parameters. Multichunk refers to parallelism across compartments. We invoke the Julia solver AutoVern9(Rodas5(autodiff = false)) with a tolerance of $1{\times}10^{-5}$, reflecting the convergence tolerance. For the second-order adjoint method, the ForwardDiffOverAdjoint(QuadratureAdjoint(autodiff = false) solver option was used.

| First-order Methods | $t_{end} = 10$ | $t_{end} = 100$ | $t_{end} = 1000$ |
|---|---|---|---|
| Complex Perturbation | $3.977 \times 10^2$ | $2.195 \times 10^3$ | $2.332 \times 10^4$ |
| Complex Perturbation Multithread | $3.661 \times 10^2$ | $2.480 \times 10^3$ | $2.330 \times 10^4$ |
| Forward | $5.404 \times 10^2$ | $2.597 \times 10^3$ | $2.505 \times 10^4$ |
| Forward Multithread | $4.527 \times 10^2$ | $2.601 \times 10^3$ | $2.336 \times 10^4$ |
| Forward Multichunk | $3.759 \times 10^2$ | $1.661 \times 10^3$ | $1.417 \times 10^4$ |
| Forward Multichunk Multithread | $2.699 \times 10^2$ | $1.352 \times 10^3$ | $1.215 \times 10^4$ |
| Adjoint | $6.118 \times 10^4$ | $5.097 \times 10^6$ | $7.825 \times 10^8$ |
| **Second-order Methods** | $t_{end} = 10$ | $t_{end} = 100$ | $t_{end} = 1000$ |
| Complex Perturbation | $2.039 \times 10^3$ | $1.245 \times 10^4$ | $1.469 \times 10^5$ |
| Complex Perturbation Multithread | $2.123 \times 10^3$ | $1.206 \times 10^4$ | $1.573 \times 10^5$ |
| Forward | $2.749 \times 10^3$ | $1.239 \times 10^4$ | $1.376 \times 10^5$ |
| Forward Multithread | $1.737 \times 10^3$ | $1.011 \times 10^4$ | $1.735 \times 10^5$ |
| Forward Multichunk | $1.097 \times 10^3$ | $4.475 \times 10^3$ | $5.382 \times 10^4$ |
| Forward Multichunk Multithread | $7.135 \times 10^2$ | $3.181 \times 10^3$ | $3.967 \times 10^4$ |
| Adjoint | $2.048 \times 10^4$ | $2.795 \times 10^5$ | $7.536 \times 10^6$ |

**Table 4. Computational time (μs) for the ROBER ODE model.** Parameters match those previously introduced in this manuscript. Multithread refers to parallelism across parameters. Multichunk refers to parallelism across compartments. We invoke the Julia solver Rodas4(autodiff = false) with a tolerance of $1\times10^{-7}$, reflecting the convergence tolerance. Second-order adjoint method not included at t = 1000 due to time constraints. For the second-order adjoint method, the ForwardDiffOverAdjoint(QuadratureAdjoint(autodiff = false) solver option was used.

| First-order Methods | $t_{end}$ = 10 | $t_{end}$ = 100 | $t_{end}$ = 1000 |
|---|---|---|---|
| Complex Perturbation | $2.475 \times 10^3$ | $4.111 \times 10^3$ | $4.117 \times 10^3$ |
| Complex Perturbation Multithread | $1.549 \times 10^3$ | $2.600 \times 10^3$ | $5.016 \times 10^3$ |
| Forward | $3.029 \times 10^3$ | $4.544 \times 10^3$ | $8.271 \times 10^3$ |
| Forward Multithread | $1.726 \times 10^3$ | $2.905 \times 10^3$ | $4.766 \times 10^3$ |
| Forward Multichunk | $1.471 \times 10^3$ | $2.422 \times 10^3$ | $4.113 \times 10^3$ |
| Forward Multichunk Multithread | $1.343 \times 10^3$ | $2.442 \times 10^3$ | $3.902 \times 10^3$ |
| Adjoint | $1.456 \times 10^8$ | $2.656 \times 10^9$ | $2.069 \times 10^{10}$ |
| **Second-order Methods** | $t_{end} = 10$ | $t_{end} = 100$ | $t_{end} = 1000$ |
| Complex Perturbation | $7.985 \times 10^3$ | $1.250 \times 10^4$ | $2.306 \times 10^4$ |
| Complex Perturbation Multithread | $5.157 \times 10^3$ | $8.868 \times 10^3$ | $1.763 \times 10^4$ |
| Forward | $7.422 \times 10^3$ | $1.101 \times 10^4$ | $2.291 \times 10^4$ |
| Forward Multithread | $4.062 \times 10^3$ | $6.131 \times 10^3$ | $1.403 \times 10^4$ |
| Forward Multichunk | $1.420 \times 10^3$ | $2.157 \times 10^3$ | $3.655 \times 10^3$ |
| Forward Multichunk Multithread | $1.439 \times 10^3$ | $2.159 \times 10^3$ | $3.552 \times 10^3$ |
| Adjoint | $3.669 \times 10^7$ | $7.388 \times 10^8$ | – |

Tables 6 and 7 compare the computational speeds of the different methods for the stochastic SIR and branching process models. As expected for the stochastic SIR model, computational speed varies roughly quadratically with the number $N$ of individuals in the system. In the stochastic SIR model, the complex perturbation method proves to be twice as fast as the manual differentiation of (Eq 10) and (Eq 8) because the latter requires a larger number of individual computations. For the branching process model however, this trend reverses since manual

**Table 5. Computational time (μs) for the MCC ODE model.** Parameters match those previously introduced in this manuscript. Multithread refers to parallelism across parameters. Multichunk refers to parallelism across compartments. We invoke the Julia solver AutoVern9(Rodas5(autodiff = false)) with a tolerance of $1\times10^{-5}$, reflecting the convergence tolerance. For the second-order adjoint method, the ForwardDiffOverAdjoint(QuadratureAdjoint(autodiff = false) solver option was used.

| First-order Methods | $t_{end}$ = 10 | $t_{end}$ = 100 | $t_{end}$ = 1000 |
|---|---|---|---|
| Complex Perturbation | $2.952 \times 10^3$ | $2.588 \times 10^4$ | $8.50 \times 10^4$ |
| Complex Perturbation Multithread | $1.806 \times 10^3$ | $1.521 \times 10^4$ | $4.612 \times 10^4$ |
| Forward | $2.758 \times 10^3$ | $1.527 \times 10^4$ | $7.741 \times 10^4$ |
| Forward Multithread | $2.147 \times 10^3$ | $1.524 \times 10^4$ | $4.646 \times 10^4$ |
| Forward Multichunk | $1.071 \times 10^3$ | $6.806 \times 10^4$ | $1.709 \times 10^4$ |
| Forward Multichunk Multithread | $8.038 \times 10^2$ | $5.494 \times 10^3$ | $1.325 \times 10^4$ |
| Adjoint | $3.601 \times 10^5$ | $3.029 \times 10^7$ | $3.332 \times 10^9$ |
| **Second-order Methods** | $t_{end} = 10$ | $t_{end} = 100$ | $t_{end} = 1000$ |
| Complex Perturbation | $3.336 \times 10^4$ | $4.457 \times 10^5$ | $1.262 \times 10^6$ |
| Complex Perturbation Multithread | $3.969 \times 10^4$ | $2.969 \times 10^4$ | $1.198 \times 10^6$ |
| Forward | $6.331 \times 10^4$ | $5.213 \times 10^5$ | $1.383 \times 10^6$ |
| Forward Multithread | $3.465 \times 10^4$ | $3.445 \times 10^5$ | $1.116 \times 10^6$ |
| Forward Multichunk | $2.257 \times 10^4$ | $1.392 \times 10^5$ | $2.886 \times 10^5$ |
| Forward Multichunk Multithread | $1.544 \times 10^4$ | $8.824 \times 10^4$ | $2.007 \times 10^5$ |
| Adjoint | $6.589 \times 10^5$ | $2.041 \times 10^7$ | $7.388 \times 10^8$ |

**Table 6. Computational time (μs) in the stochastic SIR model.** Model parameters match those previously described in this manuscript. Manual differentiation relies on differentiating Eq 7 and Eq 6 for the stochastic SIR model.

| $\partial M/\partial\delta$ | N = 10 | N = 100 | N = 1000 |
|---|---|---|---|
| Complex Perturbation | $1.90 \times 10^1$ | $1.634 \times 10^3$ | $1.975 \times 10^5$ |
| Manual Differentiation | $3.80 \times 10^1$ | $3.879 \times 10^3$ | $4.925 \times 10^5$ |
| $\partial T/\partial\delta$ | N = 10 | N = 100 | N = 1000 |
| Complex Perturbation | $1.86 \times 10^1$ | $1.620 \times 10^3$ | $2.006 \times 10^5$ |
| Manual Differentiation | $3.45 \times 10^1$ | $4.213 \times 10^3$ | $4.875 \times 10^5$ |

**Table 7. Computational time (μs) for the branching process model.** Model parameters are generated randomly on the range $\beta \in [0.05,0.16]$, $\delta \in [0.05,0.19]$, and $\lambda \in [0.0003,0.00046]$. Manual differentiation relies on differentiating Eq 11 and Eq 9 for the branching process model.

| $\partial A/\partial\delta_1$ | N = 10 | N = 100 | N = 1000 |
|---|---|---|---|
| Complex Perturbation | $2.43 \times 10^3$ | $2.33 \times 10^5$ | $1.36 \times 10^8$ |
| Manual Differentiation | $1.08 \times 10^1$ | $3.75 \times 10^4$ | $4.97 \times 10^5$ |
| Forward | $1.79 \times 10^2$ | $2.96 \times 10^5$ | – |
| Forward Multichunk | $2.85 \times 10^1$ | $1.32 \times 10^5$ | $1.39 \times 10^9$ |
| $\partial e/\partial\delta_1$ | N = 10 | N = 100 | N = 1000 |
| Complex Perturbation | $1.04 \times 10^3$ | $3.44 \times 10^4$ | $4.25 \times 10^6$ |
| Manual Differentiation | $4.26 \times 10^2$ | $4.90 \times 10^4$ | $3.19 \times 10^6$ |
| Forward | $1.03 \times 10^4$ | $1.33 \times 10^6$ | – |
| Forward Multichunk | $1.23 \times 10^3$ | $1.12 \times 10^6$ | $2.27 \times 10^9$ |

**Table 8. Prediction error results for ODE models.** Derivatives are calculated with the forward method. All predictions are for a 10% change in parameter. Parameters match those previously introduced in this manuscript.

| ODE Models | $t_{end} = 10$ | $t_{end} = 100$ | $t_{end} = 1000$ |
|---|---|---|---|
| SIR First Order | $3.370 \times 10^1$ | $2.444 \times 10^6$ | $2.524 \times 10^5$ |
| SIR Second Order | $8.208 \times 10^0$ | $2.299 \times 10^6$ | $2.303 \times 10^5$ |
| CARRGO First Order | $6.195 \times 10^{-1}$ | $2.801 \times 10^3$ | $1.465 \times 10^5$ |
| CARRGO Second Order | $1.116 \times 10^{-2}$ | $4.956 \times 10^2$ | $4.217 \times 10^4$ |
| ROBER First Order | $3.205 \times 10^{-5}$ | $3.588 \times 10^{-5}$ | $1.837 \times 10^{-5}$ |
| ROBER Second Order | $1.753 \times 10^{-6}$ | $2.201 \times 10^{-6}$ | $1.039 \times 10^{-6}$ |
| MCC First Order | $3.467 \times 10^{-4}$ | $7.556 \times 10^{-5}$ | $1.542 \times 10^{-4}$ |
| MCC Second Order | $1.268 \times 10^{-4}$ | $1.922 \times 10^{-5}$ | $3.918 \times 10^{-5}$ |

**Table 9. Prediction error results for the stochastic SIR model.** Derivatives are calculated with the complex perturbation. All predictions are for a 10% change in parameter. Parameters match those previously introduced in this manuscript.

| Stochastic SIR Model | N = 10 | N = 100 | N = 1000 |
|---|---|---|---|
| Total Number Infected ($M$) from $\eta$ | $2.322 \times 10^{-3}$ | $2.009 \times 10^{-2}$ | $1.241 \times 10^1$ |
| Total Number Infected ($M$) from $\delta$ | $4.456 \times 10^{-3}$ | $2.586 \times 10^{-2}$ | $3.670 \times 10^{-2}$ |
| Time to Extinction ($T$) from $\eta$ | $1.601 \times 10^{-3}$ | $6.715 \times 10^{-3}$ | $4.046 \times 10^{-3}$ |
| Time to Extinction ($T$) from $\delta$ | $2.074 \times 10^{-1}$ | $1.599 \times 10^{-1}$ | $4.811 \times 10^{-2}$ |

**Table 10. Prediction error results for the branching process model.** Derivatives are calculated with the complex perturbation method. Parameters are generated randomly on the range β in [0.05,0.16], λ in [0.0003,0.00046], and δ = β +.03 for calculation of a sub-critical system (A) and δ = β−.03 for calculation of a super-critical system (e).

| Branching Process Model | $N = 10$ | $N = 100$ | $N = 1000$ |
|---|---|---|---|
| Total Number Infected ($A$) from $\beta_1$ | $3.025 \times 10^{-2}$ | $7.020 \times 10^{-6}$ | $7.234 \times 10^{-9}$ |
| Total Number Infected ($A$) from $\delta_1$ | $5.036 \times 10^{-2}$ | $8.734 \times 10^{-5}$ | $1.348 \times 10^{-7}$ |
| Total Number Infected ($A$) from $\lambda_{1,1}$ | $2.402 \times 10^{-4}$ | $4.229 \times 10^{-6}$ | $4.152 \times 10^{-8}$ |
| Extinction Probability ($e$) from $\beta_1$ | $1.119 \times 10^{-4}$ | $3.476 \times 10^{-7}$ | $7.257 \times 10^{-10}$ |
| Extinction Probability ($e$) from $\delta_1$ | $7.682 \times 10^{-4}$ | $3.776 \times 10^{-6}$ | $9.424 \times 10^{-9}$ |
| Extinction Probability ($e$) from $\lambda_{1,1}$ | $5.123 \times 10^{-5}$ | $3.044 \times 10^{-6}$ | $5.800 \times 10^{-8}$ |

differentiation relies on fast linear algebra rather than iteration and avoids the overhead of complex arithmetic. The derivatives of **A** are matrix equations, and in this case forward mode differentiation even without chunking performs as well as the complex perturbation method, although it does not scale as well to larger systems ($N = 1000$). However, in the case of the derivatives of **e**, which are calculated using recursion, neither implementation of forward mode differentiation can be computed as quickly as the complex perturbation method, and this difference increases with the size of the system. Other evidence not shown suggests that the complex perturbation method can reliably evaluate sensitivities where solutions depend on linear algebra and/or recurrence relations. In summary, unless derivatives are quite complicated, manual differentiation is generally more computationally efficient than either the complex perturbation method or the forward method. In computing second derivatives, we expect the tables will be turned. To their credit, the forward and complex perturbation methods do not require formulating derivatives analytically in advance and are consequentially easier to implement.

## 4.4 Prediction error

In general, prediction error measures how well the first and second-order sensitivities capture the change in behavior of a model. Since we have previously shown that the various methods for computing differential sensitivity yield nearly the same results, prediction error is a good metric for determining the value of differential sensitivity in a particular model. We measure prediction error by the Euclidean norms

$$\text{err}_1 = \|f(\mathbf{x} + \mathbf{v}) - f(\mathbf{x}) - df(\mathbf{x})\mathbf{v}\|$$

$$\text{err}_2 = \|f(\mathbf{x} + \mathbf{v}) - f(\mathbf{x}) - df(\mathbf{x})\mathbf{v} - \frac{1}{2}\mathbf{v}^t d^2f(\mathbf{x})\mathbf{v}\|.$$

Other norms, such as the $\ell_1$ and $\ell_\infty$ norms, yield similar results. In the ODE models, $f(\mathbf{x})$ denotes a matrix trajectory so the Frobenius norm applies. To capture proportional prediction errors, we normalize all vector outputs by their length and all matrix outputs by the square of their length.

Prediction accuracy varies widely between models and even between parameters. As we expect however, second-order approximations are more accurate in prediction. Tables 8, 9 and 10 record prediction errors for each model. For the ODE systems, we see that stiffness highlights the added value of the second-order approximations. In the ROBER and CARRGO models, the second-order approximations have an order of magnitude less prediction error than the first-order approximations. However, stiffness does not appear to impact how the prediction errors grow over time. The ROBER and MCC models do not suffer from increased errors per time point after longer prediction intervals.

     

In the stochastic SIR model, prediction error does not seem to be compounded at all; in fact, the error per value calculated decreases in the case of $\frac{\partial M}{\partial \eta}$. In the case of branching processes with many types $N$ and large parameter sets, it is inadvisable to compare prediction accuracy across system sizes. However, we can conclude from these results that at least the prediction error does not compound as $N$ increases. Furthermore, prediction accuracy for the branching process models appears to vary dramatically depending on the parameter in question.

## 5 Discussion

Our purpose throughout has been to demonstrate the ease and utility of incorporating differential sensitivity analysis in dynamical modeling. Because models are always approximate, and parameters are measured imprecisely, uncertainty plagues virtually all dynamical models. While improving models is incremental and domain specific, sensitivity analysis provides a handle on local parameter uncertainty across models.

Of the methods mentioned in this text, the adjoint method, forward method, and complex perturbation methods all require that the functions defining a model be differentiable in the underlying parameters. While the complex perturbation method has the additional requirement that these functions be complex analytic, it is the only method explored in this manuscript that can be extended to discrete stochastic models in addition to ODE systems. For the modeler who prefers a one size fits all approach, or who prefers to prioritize ease of implementation, we argue that the complex perturbation method should be the method of choice. In addition to its wide range of applicability, the complex perturbation method can be easily multi-threaded and requires only implementation of the component functions of the model. In contrast to the second-order complex perturbation method, forward differentiation slows dramatically in calculating a Hessian directly. It becomes competitive if one calculates the gradient of the gradient. The gradient of the gradient method is not always available natively and usually must be implemented separately as we have done in the current manuscript. Crucially, implementing a specialized forward mode method was possible due to the underlying automatic differentiation software's flexibility and support for composition.

In situations demanding computational speed, our results suggest that choosing a method tailored to a model may be pertinent. In the case of stochastic models, manually differentiating and applying the chain rule must be balanced against the complex perturbation method, which requires less effort up front but longer processing after the derivatives have been determined. For ODE systems models, forward mode is the most computationally efficient when multichunking is available. If multichunking is not available, then the complex perturbation method has comparable speed to the forward method when run with the same tolerance. In maximizing computational efficiency, it is important to note that the use of automatic differentiation tools may require more user input for algorithm selection or multi-threading implementation. Choice of software is critical as well; not all software packages with automatic forward differentiation support chunking as implemented in the ForwardDiff.jl package and that so dramatically improves the computational efficiency of this method.

There are additional challenges to computing model sensitivity that we do not address. For example, not all models use functions that are differentiable in their parameters. Additionally, models may be differentiable yet extremely stiff, in which case the computational time for each sensitivity method discussed here will suffer disproportionally as the number of parameters grows. Furthermore, assessing global parameter sensitivity is more challenging. It can be attacked by techniques such as Latin square hypercube sampling or Sobel quasi-random sampling, but these become infeasible in high dimensions [34]. Given the availability of

appropriate software, differential sensitivity is computationally feasible, even for high-dimensional systems.

In the case of stochastic models, traditional methods require costly and inaccurate simulation over a bundle of parameter values. Differential sensitivity is often out of the question. Current automatic differentiation systems, such as PyTorch, Zygote and ForwardDiff, treat generated random numbers as constants, and thus are not reliable methods for use in calculating differential sensitivity of model outcomes that depend on these random variables. This limits the ability of researchers to understand a biological system and how it responds to parameter changes. If a system index such as a mean, variance, extinction probability, or extinction time can be computed by a reasonable algorithm, then differential parameter sensitivity analysis can be undertaken. We have indicated in a handful of examples how this can be accomplished.

In summary, across many models representative of computational biology, we have reached the following conclusions:

a.  Forward mode, adjoint, and complex perturbation sensitivity methods all converge to the same differential sensitivity values in non-stiff models, thus offering the same level of accuracy for all methods. For stiff models, forward mode and complex perturbation methods converge but adjoint sensitivity struggles and does not achieve convergence for realistic tolerance parameters.

b.  Chunked forward mode automatic differentiation and forward mode sensitivity analysis tend to be the most computationally efficient on the tested models.

c.  The complex perturbation methods described in this manuscript are competitive and often outperform the unchunked version of forward mode automatic differentiation, while being less sensitive to stiffness than the adjoint methods.

d.  Shared memory multi-threading of the complex perturbation and forward mode automatic differentiation methods provides a performance gain but only in high-dimensional systems.

e.  Forward mode automatic differentiation method requires that each step of a calculation is differentiable. This renders it unusable for calculating the derivative of ensemble means of discrete state models, such as birth-death processes. For these cases, the complex perturbation method outperforms manual differentiation.

f.  The complex perturbation method is competitive with automatic differentiation methods in accuracy, is more straightforward to implement, and can be applied to a wider variety of methods.

These conclusions are tentative but supported by our limited number of biological case studies.

We note that the performance differences may change depending on the efficiency of the implementations. The Julia DifferentialEquations.jl library and its DiffEqSensitivity.jl package have been shown to be highly efficient, outperforming other libraries in both equation solving and derivative calculations in Python, MATLAB, C, and Fortran [19,33]. Details on the current state of performance can be found at https://github.com/SciML/SciMLBenchmarks.jl.

The automatic differentiation implementations in machine learning libraries optimize array operations much more than scalar operations. This can work to the detriment of forward mode AD. MATLAB or Python style vectorization improves the performance of forward mode AD sensitivity analysis by reducing interpreter overhead. Therefore, our conclusions

serve as guidelines for the case where all implementations are well-optimized. For programming languages with high overheads or without compile-time optimization of the automatic differentiation passes, the balance in efficiency shifts more favorably towards the complex perturbation method.

One last point worth making is on the coding effort required by the various methods. Both automatic differentiation and the complex perturbation method have comparable accuracy when applied to systems of ODEs, with automatic differentiation having the advantage in speed when it is implemented with the additional level of parallelization provided by chunking. However, the complex perturbation method can easily be generalized to other kinds of objective functions and may be more straightforward to implement for those less sophisticated in computer science. While automatic differentiation is the basis of many large scientific packages, the code required for the complex perturbation methods is fully contained within this manuscript and is easily transferable to other programming languages with similar dispatching on complex numbers. This hard to measure benefit should not be ignored by practicing biologists who simply wish to quickly arrive at reasonably fast code.

## Supporting information

**S1 Appendix. Derivation of Second Derivative Complex Perturbation Method.**
(DOCX)

**S2 Appendix. Additional Models.**
(DOCX)

**S3 Appendix. Sensitivity of Linear Systems.**
(DOCX)

## Acknowledgments

We wish to thank Chris Elrod for assistance in adding multithreading to parts of our software. We wish to thank Janet Sinsheimer, Mary Sehl, Xiang Ji, and Jason Xu for helpful comments on the manuscript and biological applications.

The U.S. Government is authorized to reproduce and distribute reprints for Government purposes notwithstanding any copyright notation herein.

## Author Contributions

**Conceptualization:** Kenneth Lange.

**Methodology:** Rachel Mester, Chris Rackauckas, Kenneth Lange.

**Software:** Rachel Mester, Alfonso Landeros, Chris Rackauckas.

**Visualization:** Rachel Mester.

**Writing – original draft:** Rachel Mester, Kenneth Lange.

**Writing – review & editing:** Alfonso Landeros, Chris Rackauckas.

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
