## [Decision Letter · Decision Letter 0]

10 Dec 2021

Dear Dr. Lange,

Thank you very much for submitting your manuscript "Differential Methods for Assessing Sensitivity in Biological Models" for consideration at PLOS Computational Biology.

As with all papers reviewed by the journal, your manuscript was reviewed by members of the editorial board and by several independent reviewers. In light of the reviews (below this email), we would like to invite the resubmission of a significantly-revised version that takes into account the reviewers' comments.

Especially ensure to highlight why the readers of PLoS Computational biology are the right audience for this manuscript.  

We cannot make any decision about publication until we have seen the revised manuscript and your response to the reviewers' comments. Your revised manuscript is also likely to be sent to reviewers for further evaluation.

Sincerely,

Attila Csikász-Nagy

Associate Editor

PLOS Computational Biology

Mark Alber

Deputy Editor

PLOS Computational Biology

Reviewer's Responses to Questions

**Comments to the Authors:**

Reviewer #1: This paper is a useful overview and comparison of sensitivity analysis methods for deterministic and stochastic biological models. I fully agree that (time dependent) sensitivity is fundamental in parameter estimation, experiment design, control input selection or measurement setup. Moreover, in many cases, researchers tend to apply less efficient (e.g., simulation based) methods, although there are more powerful and also more informative alternatives. In this respect this paper has a really high tutorial value. The paper is generally well-written and it successfully balances between the necessary formal notations and textual explanation. The presented Julia codes are useful and easy to understand.

Specific comments:

- In the Introduction part, it would be even more motivating to write about how sensitivity may determine the quality of parameter estimation results, e.g. in the context of structural and/or practical identifiability.

- I'd suggest structuring Section 2 into subsections corresponding to the different methods described.

- There seems to be a typo in the inline formula of the first order Taylor approximation in line 75, page 3.

- I'd suggest changing superscript t for the notation of transposed to someting else, e.g. to superscript T. Otherwise it can be confused with the time argument.

- Please characterize the parameter dependence of matrix $A$ on $\\beta$ in the beginning of Section 2.

- What is 'original' in Fig. 4?

- In Table 3, the label of the 3rd sub-column of 'Time' should be $t_{end}=1000$

- Similarly, in Table 4, the 3rd sub-column of 'Time' should be $N=1000$

- Does irreducibility of the branching process mean that the transition graph is strongly connected?

Reviewer #2: This work showcases the use of differential sensitivity analysis for a small set of biologically relevant models including ODEs and markov chains. As for as I can tell the only novel result in this paper is equation (4) where the authors make use of the complex plane to approximate hessians. The main results focus of performance and complexity benchmarks between this analytic approach, forward-mode autodiff and finite differences. While such studies are important in machine learning literature, they are of little interest to a biological audience and hence not appropriate for publication in this journal.

The overall quality of figures is poor, using the default styling of the Julia library Plots.jl and paying little attention to matching notation between code, figure legends and math in the main text. The narrative reads more like a tutorial and my recommendation would be to restructure this work as such, perhaps submitting to next year's JuliaCon workshops. I've attached the main text with comments on how to improve the text.

Reviewer #3: This work compares different numerical methods for computing the local sensitivities of parameter-dependent ODEs. The authors implement three classical methods (forward, adjoint, automatic differentiation) and a newer complex analytic method that is described in this manuscript (although it is not clear how much of this fourth method is a unique contribution of the current manuscript). Helpful Julia code is provided to illustrate each approach (although this code sometimes disrupts the flow of the manuscript and could be replaced with pseudocode and moved to an appendix or supplemental material). The numerical examples are based on ODEs and stochastic models taken from oncology and epidemiology. By comparing the accuracy and computational cost across methods, the authors hoped to provide practical guidance for computational biologists to decide which numerical method to implement for their sensitivity analysis needs. The goal of the manuscript seemed to be to provide insights into the complex tradeoff between accessibility, speed, and accuracy of these four methods as they are applied to dynamic models in biology. This goal is extremely worthwhile and exciting. If achieved, a careful exposition on the advantages and disadvantages of different methods for sensitivity analysis for different types of deterministic and stochastic biological models would be very helpful to

a wide audience of computational scientists. However, as it is currently written, the manuscript falls very short of these goals, and substantial improvements both in clarity of presentation and rigor of the numerical analyses would be needed for the manuscript to be acceptable for publication. More detailed comments are given below.

Major comments

1) The metrics used to quantify accuracy do not address the accuracy in the calculation of the local sensitivity (section 4, lines 613-614). Rather, it appears that these metrics quantify how well the local behavior of the model can be approximated by a linear or quadratic Taylor approximation. Still, it leaves open whether the differentials themselves are well-approximated by the numerical methods. The latter seems to be more interesting to the users of these numerical methods, who may have chosen to use a 1st or 2nd order sensitivity analysis and wish to know which is more accurate/fast/accessible.

2) In practice, users tend to face the complex tradeoff between different algorithmic parameters, such as the tolerance of the ODE solver, the step size of the finite-difference method versus computational budget. The provided numerical benchmarks do not seem to give any insights into these important concerns. Please make more explicit what ODE solver was used and what were the values of the error tolerances. Could the users afford to relax some of those error tolerances and still achieve acceptable accuracy? Perhaps a graph showing the dependence between (complex) finite-difference step size and/or ODE solver tolerance versus accuracy will be helpful here.

3) Parameter sensitivity analysis is a well-established field that is heavily used in computational biology for the interpretation and design of experiments. Unfortunately, as this paper is written, the introduction and discussion sections do not provide a thorough enough discussion of previous works to set the stage or motivate new efforts this important topic. Instead, the introduction reads more like a collection of anecdotes and generalizations, and the reference list is incredibly short for such a well-studied topic, lacking references from key players in the field such as M. Stumpf, DE Kirschner, F. Doyle, M. Khammash. A more thorough and scientifically documented introduction that better explains the uses for, and limitations of, existing sensitivity analyses would help place the current work in a far more compelling context that would motivate readers to appreciate the importance of the manuscript’s results.

4) The writing of this document does not provide enough discussion of the advantages and limitations of the obtained results, and the manuscript does not sufficiently describe the biological contexts in which these results will, or will not, be important.

5) There are problems with, or missing information in, many figures: Figure 2 lacks a y-axis label. Figure 1 is cropped to obscure the colorbar and is missing a label for the colorbar. Several figure captions are not provided or are not sufficiently descriptive. Fonts on several figures are too small to read for both labels and axes values, and some of the plot lines are too thin to see very well. Please consider carefully to improve the quality and format of the figures -- this could increase the overall quality of the final document.

6) The discussion of the advantage of multithreading on the MCC model (Lines 664-665 main text) is not convincing and seems to contradict the CPU times reported in the last sub-table of Table 2. In particular, the multi-threaded versions of both first and second-order methods seem to improve only slightly for the case t_end=10 and have mixed outcomes for t_end=100 and t_end=1000. Moreover, the specifics of how multithreading is accomplished are not sufficiently clear in the manuscript.

Minor Comments

7) The insertion of the Julia codes in the main text is somewhat distracting, especially for readers who may be more experienced in other coding languages. We would recommend moving the specific Julia codes to appendices and GitHub links, and then replace the algorithms in the main text with language-agnostic pseudo code.

8) Multithreading is mentioned in the manuscript, but the example Julia routines did not seem to have any thread directives. How much complexity would be added to the code if we wish to implement the multi-threaded versions of the methods?

9) Please provide more details on how parallelism is done for the implementations. For example, with the complex-analytic method, is each function evaluation distributed among threads, but is the computation of the ODE at each parameter perturbation done sequentially?

10) In Table 2, it is not clear what the difference between "Parallel FD.jl" and "Multi-Threaded FD.jl" is.

11) In most cases shown in Table 2, it appears that the "Multi-Threaded Analytic" version does not improve, and even slightly worsen, computational time compared to "Analytic." If the computations of the gradient entries are done in parallel, I would expect something close to linear scalability. Is there any explanation for this?

12) In Table 5 of Appendix C, what is the difference between "Parallel FD.jl Gradient" and "Parallel FD.jl Multi-Threaded Gradient"? Are there other parallelisms other than multithreading?

13) Appendix B. Mammalian Cell Cycle Model. The authors cite reference 23 as the source of this model. But a quick inspection of the reference reflects a model with 39 ODEs versus the 11 equations in this text. Also, the initial condition values reported in this manuscript are different to the initial conditions reported on the original document. Is the model presented here a simplification of the one presented by Gerard and Goldbetor? If this is the case, why are the authors not explaining these changes to the original model? How are these modifications affecting the original model?

14) Some equations are numbered but others are not? The manuscript would be easier to read with all equations given a number.

15) Line 254, page 12: please provide the citation.

16) There seem to be typos in Table 2 where, for example, "t_end=100" is repeated twice. Perhaps the third column should be "t_end=1000"? Same with "N=100" in Table 3.

17) To improve readability, please provide more information in the captions of the tables, explaining how the methods are mapped to the numerical schemes in the main text. For example, what does "Parallel FD.jl" in Table 2 correspond to?

18) Page 3, line 56 – change "These optimization techniques depend heavily on gradients and and Hessians with respect to parameters." to "These optimization techniques depend heavily on gradients and Hessians with respect to parameters."

19) Please hyphenate two-word adjectives before a noun.

20) In the paragraph starting on line 149, the adjoint variable term \\lambda is not to defined or described to sufficient detail.

21) At or near line 72, please state explicitly that \\Beta refers to the model parameters – this will make it clearer to readers later on.

22) In section 3.3, how are the second order terms calculated? This does not appear to be included in the preceding Julia codes.

23) Please carefully check the whole document and make sure that no typos and errors exist in the document's final version.

Reviewer #4: There isn't really much to say about this paper, it's a survey of different approaches to computing time-dependent sensitivities. I think it's a useful contribution, not groundbreaking in any way but a useful survey of methods and how well they perform. As far as I could tell, there weren't any really new algorithms developed but an assessment of existing ones. I think the provision of Julia code is very useful. My only tiny issue is with the forward difference method mentioned on page 8, no one in their right mind would use that, but the authors do mention the use of the two-point method in the following paragraph. One question that authors might consider in the discussion is how well these methods perform on really stiff problems and the problem with fast changing variables where the derivatives change rapidly. Differentiation is notoriously bad in such situations, and we, for example, have to resort to Richardson's extrapolation for tough problems, which is particularly slow.

**Have the authors made all data and (if applicable) computational code underlying the findings in their manuscript fully available?**

Reviewer #1: Yes

Reviewer #2: **No: **The referenced code repository [https://github.com/rachelmester/SensitivityAnalysis] appears to be private

Reviewer #3: Yes

Reviewer #4: Yes

PLOS authors have the option to publish the peer review history of their article (what does this mean?). If published, this will include your full peer review and any attached files.

Reviewer #1: No

Reviewer #2: **Yes: **Grisha Szep

Reviewer #3: No

Reviewer #4: No
---

## [Decision Letter · Decision Letter 1]

8 Apr 2022

Dear Ms. Mester,

Thank you very much for submitting your manuscript "Differential Methods for Assessing Sensitivity in Biological Models" for consideration at PLOS Computational Biology. As with all papers reviewed by the journal, your manuscript was reviewed by members of the editorial board and by several independent reviewers. The reviewers appreciated the attention to an important topic. Based on the reviews, we are likely to accept this manuscript for publication, providing that you modify the manuscript according to the review recommendations.

Ensure that all figures are publication quality and consider suggestions by referee 2.

Sincerely,

Attila Csikász-Nagy

Associate Editor

PLOS Computational Biology

Mark Alber

Deputy Editor

PLOS Computational Biology

[LINK]

Reviewer's Responses to Questions

**Comments to the Authors:**

Reviewer #1: The manuscript has been significantly improved compared to the initial version. The authors have addressed all of the comments and clearly explained and justified the modifications. I maintain that this paper is a really useful overview of parametric sensitivity and the related computational tools for the analysis of biological systems described by dynamical models.

Reviewer #2: Although the authors have made positive changes to the manuscript with sections 1-4. The quality of figures and math is still poor. In my opinion the paper is now somewhat below acceptance threshold and therefore I request a major revision. I recommend:

- revision of all figures to meet a publication quality

- focus section 4 on fowarddiff.jl vs complex perturbation, in particular advantages of complex perturb in branching processes

- remove any excess math that does not contribute to the main narrative, use clear and consistent definitions and notations for derivatives and definitions of different types of errors in the introduction sections

I must re-iterate here that we may simply disagree, or I may have mis-interpreted the target audience for this paper. As a computational biologist I can merely give my own perspective on what aspects of this work I find valuable to the community I am acquainted with and do not wish to cause offence but am trying to help. There is genuinely interesting work here that I think can be presented more succinctly.

More specific comments can be found in the attached PDF.

Best Wishes

Reviewer #3: The manuscript is very much improved and the authors have thoroughly addressed our primary concerns in the previous version. We have a small number of minor comments, but otherwise we are satisfied with the revised manuscript:s

While we agree that it is very helpful to make the Julia code easily available to the reader, the Julia code as currently formatted is still somewhat distracting from the flow of the text. Perhaps this can be improved by placing it in a labeled box - or perhaps it could be presented in a different font — we recommend working with the journal editors to choose the most appropriate way to present this.

Some of the figure fonts (especially axes labels and tick numbers) are still too small to read - we really had to zoom in to see these. Please increase to a minimum font size of 9 or 10pt.

In Fig. 7 (left), it is not clear why the errors are so large (what do values of 10^10 mean in this context?). We wonder if the Euclidean distance definition for the error is not so informative and perhaps a Euclidean distance for the relative error may be better. I expect that the the figure would look the same (except with the x-axis numbers would be scaled), so this should not change any of the conclusions.

**Have the authors made all data and (if applicable) computational code underlying the findings in their manuscript fully available?**

Reviewer #1: None

Reviewer #2: Yes

Reviewer #3: Yes

PLOS authors have the option to publish the peer review history of their article (what does this mean?). If published, this will include your full peer review and any attached files.

Reviewer #1: No

Reviewer #2: **Yes: **Grisha Szep

Reviewer #3: No

Figure Files:

Data Requirements:

Reproducibility:

References:

---

## [Editor Report · Decision Letter 2]

12 May 2022

Dear Ms. Mester,

We are pleased to inform you that your manuscript 'Differential Methods for Assessing Sensitivity in Biological Models' has been provisionally accepted for publication in PLOS Computational Biology.

Best regards,

Attila Csikász-Nagy

Associate Editor

PLOS Computational Biology

Mark Alber

Deputy Editor

PLOS Computational Biology

---

## [Editor Report · Acceptance letter]

30 May 2022

PCOMPBIOL-D-21-01956R2 

Differential Methods for Assessing Sensitivity in Biological Models

Dear Dr Mester,

I am pleased to inform you that your manuscript has been formally accepted for publication in PLOS Computational Biology. Your manuscript is now with our production department and you will be notified of the publication date in due course.

With kind regards,

Agnes Pap
